# Heritable epigenetic changes are constrained by the dynamics of regulatory architectures

**Antony M Jose***

University of Maryland, College Park, United States

**\*For correspondence:**
amjose@umd.edu

**Competing interest:** The author declares that no competing interests exist.

**Abstract** Interacting molecules create regulatory architectures that can persist despite turnover of molecules. Although epigenetic changes occur within the context of such architectures, there is limited understanding of how they can influence the heritability of changes. Here, I develop criteria for the heritability of regulatory architectures and use quantitative simulations of interacting regulators parsed as entities, their sensors, and the sensed properties to analyze how architectures influence heritable epigenetic changes. Information contained in regulatory architectures grows rapidly with the number of interacting molecules and its transmission requires positive feedback loops. While these architectures can recover after many epigenetic perturbations, some resulting changes can become permanently heritable. Architectures that are otherwise unstable can become heritable through periodic interactions with external regulators, which suggests that mortal somatic lineages with cells that reproducibly interact with the immortal germ lineage could make a wider variety of architectures heritable. Differential inhibition of the positive feedback loops that transmit regulatory architectures across generations can explain the gene-specific differences in heritable RNA silencing observed in the nematode *Caenorhabditis elegans*. More broadly, these results provide a foundation for analyzing the inheritance of epigenetic changes within the context of the regulatory architectures implemented using diverse molecules in different living systems.

## eLife assessment

This **useful** manuscript explores conditions for epigenetic inheritance by studying the stability of simple network models to permanent and transient perturbations. A novel aspect of the study is that it unifies non-genetic inheritance phenomena across cell divisions of unicellular organisms and in the germline of multicellular organisms. However, the models studied are more a collection of vignettes of numerical studies than a systematic study, therefore the evidence presented remains **incomplete**. As a first step towards building a more systematic theoretical framework, this work will be of interest to colleagues in the field of epigenetic inheritance.

## Introduction

Patterns formed by interactions between molecules can be preserved by living systems even as the molecules change over time. For example, the localization and activity of many kinds of molecules are recreated in successive generations during comparable stages (*Jose, 2018*). These recurring patterns can change throughout development such that following the levels and/or localizations of each kind of molecule over time traces waveforms that return in phase with the similarity of form and function across generations (*Jose, 2020c*). At any time, interactions that can be used to predict future arrangements of molecules define regulatory architectures that drive change or preserve homeostasis. Such architectures that arose with the origin of life have since diversified through descent with modification

to form the many heritable regulatory architectures that are now transmitted across generations along with the genome.

Interactors that form regulatory architectures can span many scales, but descriptions at particular scales are expected to be most useful for a given experimental technique or approach (*Jose, 2020a*). For example, molecules can interact to form a complex that both provides output to and receives input from another complex, which in turn might be regulated by an organelle. Such interactions can be described as 'top-down' or 'bottom-up' based on the sequential order in which different levels of organization such as molecules, complexes, organelles, cells, tissues etc. are considered to create an explanatory hierarchy. Considering these multi-scale interaction networks in terms of entities, their sensors, and the sensed properties provides a flexible framework for analysis (*Jose, 2020b*) that can be used to progressively refine models (*Figure 1—figure supplement 1*). In these entity-sensor-property (ESP) systems, all interactors of interest can be conveniently defined as entities with some entities acting as sensors. Such sensors can cause changes (either promotion or inhibition *Figure 1—figure supplement 2a*) in the rest of the system or the environment in response to changes in particular properties of other entities. As a result, the regulatory architectures can be in different states at different times, depending on the levels of all entities/sensors (*Figure 1—figure supplement 2b*). Some interactors have compositions that change over time (e.g. biomolecular condensates with molecules in equilibrium with other dissolved molecules in the surrounding liquid *Alberti et al., 2019*). Such dynamic interactors can be included by considering them as entities whose integrity and properties depend on the properties of some other entities in the system and/or the environment (see *Krakauer et al., 2020* for a similar definition for degrees of individuality). Defined in this way, all ESP systems capture regulatory architectures that could persist over time even as the interacting entities and sensors change. For example, a gated ion channel acts as a sensor when it responds to an increase in the intracellular concentrations of an ion with a change in conformation, allowing the import of other ions from the extracellular environment. During development, such an ion channel could be replaced by another with similar properties, allowing the persistence of the regulatory relationships. Therefore, the analysis of ESP systems is a useful approach for examining heritable regulatory architectures to inform mechanistic studies that aim to explain phenomena using relationships between specific interactors (e.g. epigenetic inheritance using small RNA, chromatin, 3D genome organization, etc.).

Heritability can be considered in multiple ways. In this work, for a regulatory architecture to be heritable, all interactions between different regulators need to be preserved across generations with a non-zero level of all entities in each generation. Additional notions of heredity that are possible range from precise reproduction of the concentration and the localization of every entity to a subset of the entities being reproduced with some error while the rest keep varying from generation to generation (as illustrated in Figure 2 of *Jose, 2018*). Importantly, it is currently unclear which of these possibilities reflects heredity in real living systems.

Here, I consider the transmission of information in regulatory architectures across generational boundaries to derive principles that are applicable for the analysis of heritable epigenetic changes. Only a small number of possible regulatory architectures formed by a set of interactors are heritable. Their maintenance for many generations requires positive feedback loops. Such heritable regulatory architectures carry a vast amount of information that can quickly outstrip the information that can be stored in genomes as the number of interactors increase. Quantitative simulations of perturbations from steady state suggest that these architectures can recover after many epigenetic perturbations, but some resulting changes can become heritable. Transient perturbations reveal diagnostic differences between regulatory architectures and suggest ways to generate heritable epigenetic changes for particular architectures. Unstable architectures can become heritable through periodic interactions with external sources of regulation (e.g. somatic cells for architectures within the germline), revealing a strategy for making a wider variety of regulatory architectures heritable. Transgenerational inhibition that tunes the activity of positive feedback loops in regulatory architectures can explain the gene-specific dynamics of heritable RNA silencing observed in the nematode *Caenorhabditis elegans*.

**Table 1.** Capacity of heritable regulatory architectures to store information.

The number of heritable, regulated, and heritably regulated architectures, and the information they can store were calculated using a program that enumerates non-isomorphic weakly connected graphs that satisfy specified criteria ('Heritable_Regulatory_Architectures_1–4_entities.py').

| Entities | Heritable | Regulated | Heritably regulated | Bits of information |
|---|---|---|---|---|
| 1 | 0 | 0 | 0 | - |
| 2 | 1 | 3 | 1 | 0 |
| 3 | 7 | 96 | 25 | 4.64 |
| 4 | 125 | 19,559 | 5604 | 12.45 |

## Results

### Information in heritable regulatory architectures grows rapidly with the number of interactors

As cells divide, they need to transmit all the regulatory information that maintains homeostasis. This imperative is preserved across generations through a continuum of cell divisions in all organisms as evidenced by the similarity of form and function in successive generations. Such transmission of regulatory information across generations occurs in conjunction with the sequence information transmitted by replicating the genome during each cell division. The maximal information that can be transmitted using the genome sequence is proportional to its length ($\log_2[4].l=2l$ bits for $l$ base pairs). To determine how the maximal information transmitted by interacting molecules increases with their number (*Table 1*), the regulatory architectures that can be formed by 1–4 entities were considered. Perpetual inheritance of such regulatory architectures requires sustained production of all the interacting molecules, that is every interactor must have regulatory input that *promotes* its production to overcome dilution at every cell division and other turnover mechanisms, if any. Indeed, this requirement was fundamental for conceiving the origin of life (*Eigen, 1971*; *Gánti, 1975*; *Varela et al., 1974*; *Kauffman, 1993*) and remains necessary for its persistence. Therefore, the minimal heritable regulatory architecture (HRA) is that formed by two molecules that mutually promote each other's production (*Figure 1*, 'A'), resulting in a positive feedback loop. However, not all positive feedback loops

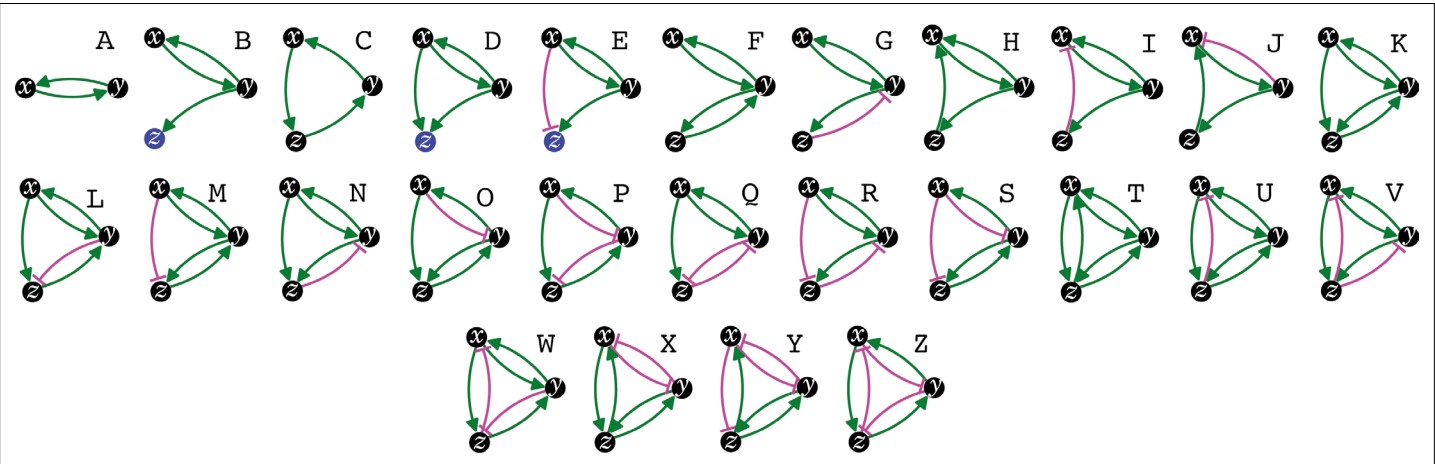

**Figure 1.** The simplest heritable regulatory architectures. Of the 99 possible regulatory architectures with fewer than four entities (see *Figure 1—figure supplement 3*), only 26 can be indefinitely heritable (**A** through **Z** with *x*, *y*, and *z* entities/sensors). Entities that act as sensors (black circles) or that do not provide any regulatory input (blue circles), or that provide positive (green arrows) or negative (magenta bar) regulatory interactions are indicated.

The online version of this article includes the following figure supplement(s) for figure 1:

**Figure supplement 1.** Entity-Sensor-Property systems provide a principled way of parsing regulators and their interactions in living systems.

**Figure supplement 2.** Illustrations of key concepts.

**Figure supplement 3.** Adding regulation to the 8 simple heritable architectures generates 99 regulated architectures, not all of which are heritable.

form HRAs. For example, positive feedback loops that promote the transient amplification of changes such as that formed by two molecules that mutually repress each other's production (*Mitrophanov and Groisman, 2008*) are not compatible with perpetual inheritance because both molecules will be eventually lost by dilution or turnover.

Distinct architectures that can be formed by a set of interacting regulators can be represented as directed graphs that are non-isomorphic and weakly connected (*Alon, 2020*: 35, 68). Imposing the need for positive regulation for heritability reveals that only 7 of the 13 possible 3-node graphs formed by 3 interactors can be HRAs and only 125 of the 199 possible 4-node graphs can be HRAs (see Methods for computation). Including either positive or negative regulation for each interaction in a HRA and then selecting only architectures that include positive regulatory input for every interactor resulted in non-isomorphic weakly connected directed graphs that represent the distinct regulatory architectures that are heritable (*Table 1*): two entities form one HRA, three form 25, and four form 5604. Thus, with four interactors, the maximal information that can be transmitted using HRAs ($\log_2[5604] \approx 12.45$ bits) surpasses that transmitted by a four base-pair long genome ($\log_2[4].4 = 8$ bits). The combinatorial growth in the numbers of HRAs with the number of interactors can thus provide vastly more capacity for storing information in larger HRAs compared to that afforded by the proportional growth in longer genomes.

## Genetic and epigenetic perturbations can generate different heritable changes

To examine how each of the 26 simplest HRAs (*Figure 1* and *Figure 1—figure supplement 3*) responds to a perturbation from steady state, ordinary differential equations that describe the rates of change of each entity in each HRA were developed (see Methods) and used to simulate steady states (*Figure 2* and *Figure 2—figure supplements 1–7*). For each regulatory architecture, positive and negative regulatory interactions (green arrow and magenta bar, respectively, in *Figure 1*) are captured as linear functions (e.g. $k_{xy}.y - k_{xz}.z$ if $x$ is positively regulated by $y$ and negatively regulated by $z$). To ensure the concentrations of all entities remain non-negative, as expected in real living systems, the equations for the rate of change are bounded to be applicable only when the values of the changing entity is greater than zero. At steady state, the concentrations of all interactors ($x_0, y_0, z_0$) remain constant because the combination of all regulatory input, which must cumulatively promote the production of each entity (with rates $k_{xy}, k_{yz}, k_{zy}$, etc), is equal to the turnover of that entity (with rates $T_x, T_y, T_z$). In principle, a genetic or non-genetic (i.e. epigenetic) perturbation could alter one or more of the following: the concentration of an entity, the strength of a regulatory link, the rate of turnover of each entity, and the polarity of an interaction. Of these, the most widely used perturbation that is easy to accomplish using current experimental techniques is reducing the concentration of an entity/ sensor (e.g. using a loss-of-function mutation, knockdown of an mRNA, degradation of a protein, etc.). Indeed, the use of genome editing (*Anzalone et al., 2020*) for removal and RNA interference (RNAi) (*Fire et al., 1998*) for reduction of an entity/sensor are common during the experimental analysis of living systems. Therefore, the impact of permanent or transient loss of an entity was compared.

To simulate genetic change, the response after removal of each entity/sensor was examined in turn for each HRA (*Supplementary file 1*, left panels in *Figure 2* and *Figure 2—figure supplements 1–7*). Deviations from unregulated turnover of the remaining entities (dotted lines, left panels in *Figure 2* and *Figure 2—figure supplements 1–7*) reveal the residual regulation and are diagnostic of different regulatory architectures. When the removed interactor was an entity with no regulatory input into the other sensors, the remaining two sensors were unaffected (e.g. left panel, loss of *z* in *Figure 2— figure supplement 1b, d and e*). Residual promotion resulted in slower decay (e.g. left panels, *y* in *Figure 2c, e and h*) and residual inhibition resulted in more rapid decay (e.g. left panels, *x* in *Figure 2g*; *y* in *Figure 2f and z* in *Figure 2c, d, e and h*). In some cases, when the remaining architecture was composed of two sensors that promote each other's production, there was continuous growth of both because their new rates of production exceed their rates of turnover (e.g. left panels, *z* in *Figure 2b* and *Figure 2—figure supplements 2 and 3*, *Figure 2—figure supplement 5b-c*, *Figure 2—figure supplement 6b-d*). In other cases, the remaining architecture resulted in slower decay of both because their new rates of production were insufficient to overcome turnover (e.g. left panels, *z* in *Figure 2*). Thus, genetic change can result in unrestrained growth or eventual decay of the remaining entities depending on the residual architecture (*Figure 1—figure supplement 2c*).

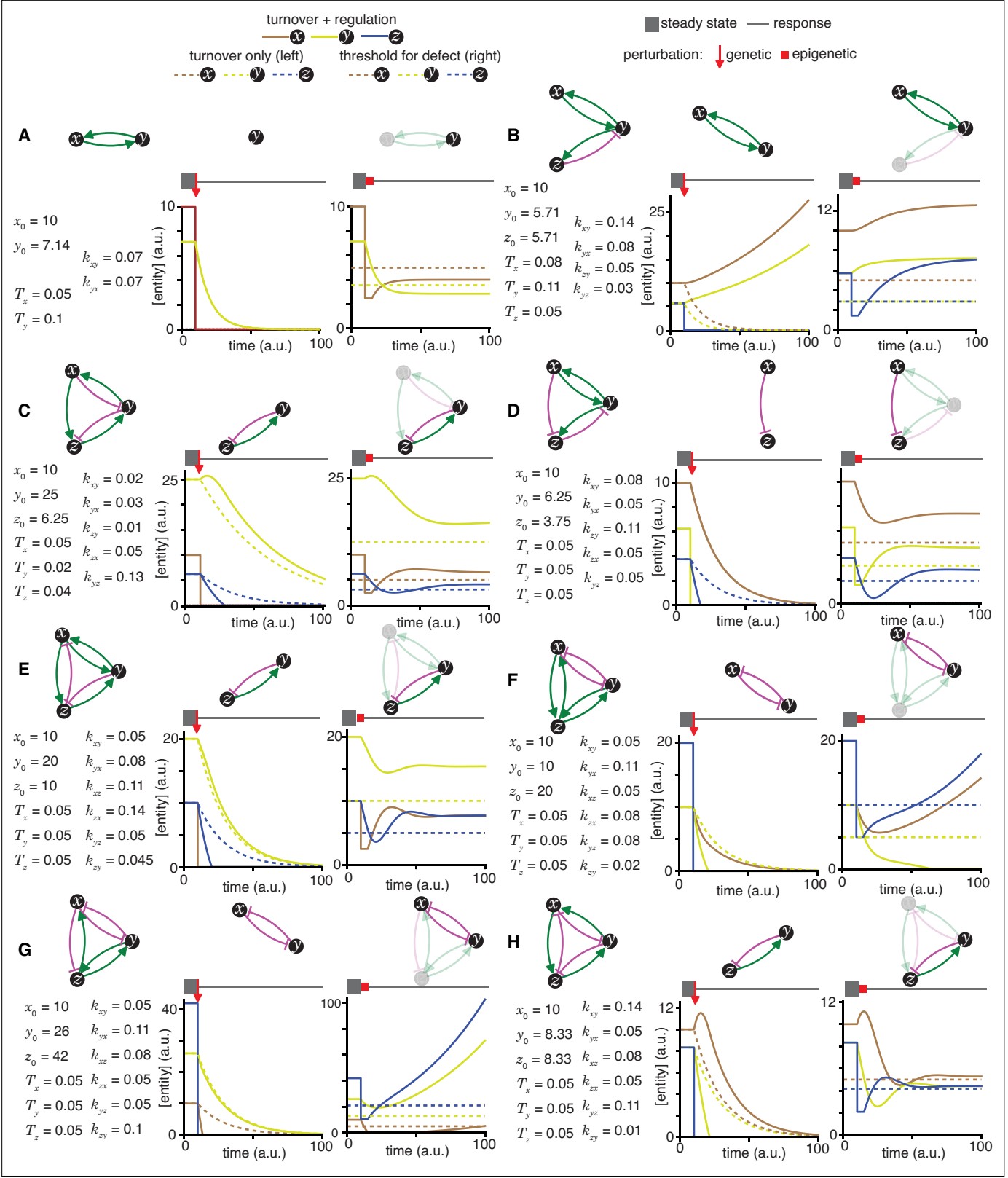

**Figure 2.** Epigenetic and genetic changes can provide complementary information about heritable regulatory architectures. (**A** to **H**) In each panel, cases where specific perturbations of architectures (*top left*) characterized by sets of parameters that support a steady state (*bottom left*) result in different outcomes for permanent or genetic (*middle*) versus transient or epigenetic (*right*) changes are illustrated. Relative concentrations of each entity during periods of steady state (thick grey line), the point of genetic change (red arrow), periods of epigenetic reduction (red bar, for a duration

*Figure 2 continued on next page*

*Figure 2 continued*

$t_p$ = 5 (a.u); with the threshold for observing a defect $d$=0.5; and an extent of perturbation beyond the threshold p=0.5), and periods of recovery after perturbation (thin grey line) are shown. Architectures are depicted as in *Figure 1* (**A, B, C, D, E, F, G** and **H** depict the heritable regulatory architectures A, G, P, R, W, X, Y and Z, respectively) with transient reductions in an entity or sensor and associated interactions depicted using lighter shades. Dotted lines indicate unregulated turnover (in *middle*) or thresholds for observing defects upon reduction in levels of an entity/sensor (in *right*).

The online version of this article includes the following figure supplement(s) for figure 2:

**Figure supplement 1.** Heritable regulatory architectures with one loop.

**Figure supplement 2.** Heritable regulatory architectures with two loops and a shared node.

**Figure supplement 3.** Heritable regulatory architectures with two loops and a shared edge.

**Figure supplement 4.** Heritable regulatory architectures with two loops, a shared node, a connecting edge, and up to one negative regulatory interaction.

**Figure supplement 5.** Heritable regulatory architectures with two loops, a shared node, a connecting edge, and two negative regulatory interaction.

**Figure supplement 6.** Heritable regulatory architectures formed by complete graphs with up to two negative regulatory interactions.

**Figure supplement 7.** Heritable regulatory architectures formed by complete graphs with three negative regulatory interactions.

To examine scenarios where epigenetic perturbations could cause heritable changes, the threshold for observing a defect was set at half of the steady-state levels (dotted line, right panels in *Figure 2* and *Figure 2—figure supplements 1–7*). RNAi can cause detectable defects that are heritable (*Fire et al., 1998*) and the conditions that promote or inhibit heritable epigenetic change after RNAi of a gene have been proposed to depend upon the regulatory architecture (*Chey and Jose, 2022*). To simulate RNAi of an entity/sensor, the response after a transient reduction of each entity/sensor to half of the threshold required for observing a defect was examined in turn for each HRA (right panels in *Figure 2* and *Figure 2—figure supplements 1–7*). The responses after this transient epigenetic perturbation were different from that after genetic perturbation (compare left and right in *Figure 2* and *Figure 2—figure supplements 1–7*), as expected. Many HRAs recovered the levels of all entities/sensors above the threshold required for detecting a defect (e.g. right panels in *Figure 2b–e , and h*). In some cases, this perturbation was sufficient to maintain the architecture but with a reduced steady-state level of all entities/sensors (e.g. reduction of *x* in *Figure 2a*, right; *Figure 2—figure supplement 1b–e*, right; *Figure 2—figure supplement 2b*, right; *Figure 2—figure supplement 5*, right). Notably, whether recovery occurs with the steady-state levels of each entity/sensor returning above or below the threshold for observing a defect depended not only on the architecture, but also on the identity of the perturbed entity/sensor (e.g. compare reduction of *x* vs. *y* in *Figure 2—figure supplement 1a*, right; *x* vs. *y* or *z* in *Figure 2—figure supplement 1b*, right; *x* vs. *y* or *z* in *Figure 2—figure supplement 1c*, right). Transient reduction of other entities/sensors below the threshold for observing defects was also observed in many cases (e.g. levels of *z* when *x* was perturbed in *Figure 2c*, right; of *z* when *y* was perturbed in *Figure 2d*, right; of *z* when *x* was perturbed in *Figure 2e*, right; of *x* and *y* when *z* was perturbed in *Figure 2h*, right). In some cases, recovery of original architectures occurred even after complete loss of one entity/sensor (e.g. after transient loss of *z* in *Figure 2g*, right; of *y* in *Figure 2—figure supplement 6*, right; of *y* in *Figure 2—figure supplement 6*, right). Such recovery from zero can be understood as re-establishment of the regulatory architecture within the duration of simulation (100 in *Figure 2* and *Figure 2—figure supplements 1–7*) and is analogous to basal activity in the absence of inducers (e.g. leaky production of LacY permease from the *lac* operon in the absence of lactose (*Robert et al., 2010*), which allows the initial import of the lactose required for activating the *lac* operon). Alternatively, such recovery can also be understood as arising from the production of the missing entity/sensor as a byproduct when the activity of the upstream regulator increases beyond a threshold. Transient perturbations were also observed to induce different architectures that can persist for many generations (e.g. transient reduction of *z* resulting in loss of *y* and mutually promoted growth of *x* and *z* in *Figure 2f*, right; also see *Figure 2—figure supplement 6* and *Figure 2—figure supplement 7*). Such continuous growth after an epigenetic change provides opportunities for achieving new steady states through dilution via cell divisions during development, potentially as part of a new cell type. Finally, some transient perturbations also led to the collapse of the entire architecture (e.g. transient perturbation of *y* in *Figure 2—figure supplement 7*, right). Thus, epigenetic change can result in unrestrained growth, eventual decay, or recovery at a new

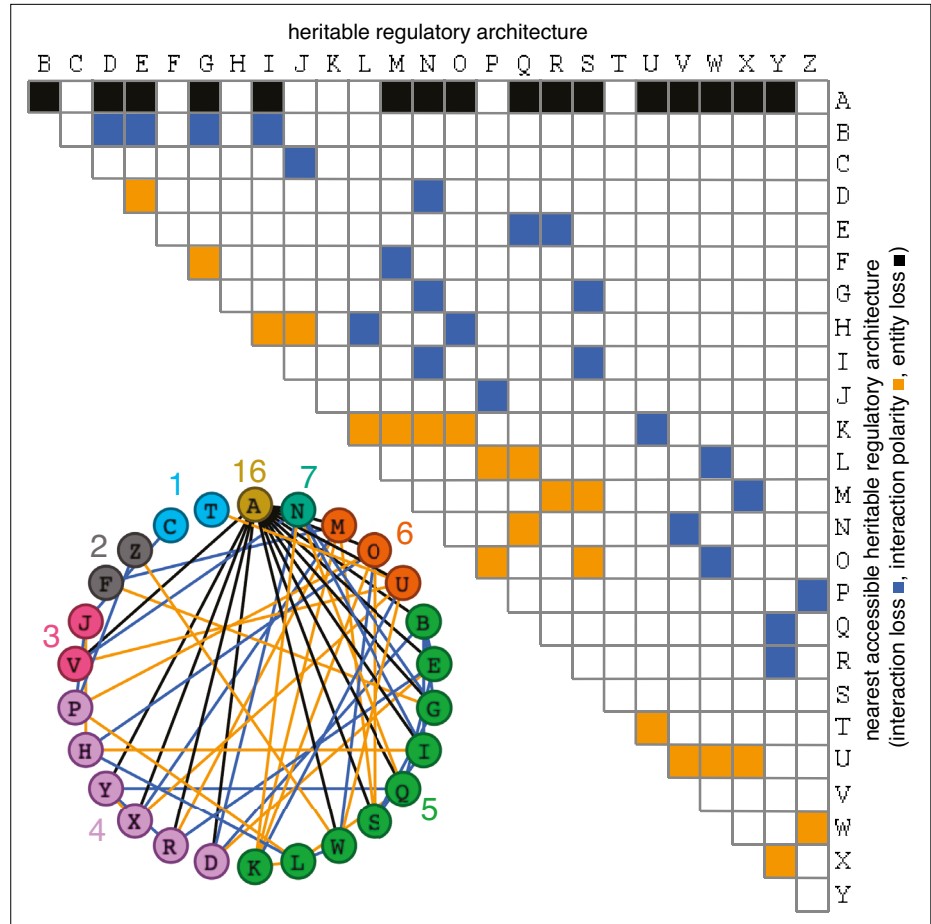

**Figure 3.** Possible conversions between the simplest heritable regulatory architectures. Table summarizing the possible changes in regulatory architecture observed after a single perturbation from steady state (blue, loss of a regulatory interaction; orange, change in the polarity of a regulatory interaction; black, either change in regulatory interaction and/or loss of an entity). For example, Z can arise from P through the loss of a regulatory interaction or from W through a change in the polarity of a regulatory interaction. *Bottom left*, Network diagram summarizing possible changes arranged clockwise by frequency of change to the HRA (color-matched numbers). Edges (black, blue, or orange) are colored as in table and nodes are colored according to number of adjacent HRAs.

steady state level of the remaining entities depending on the residual architecture and the nature of the perturbation (*Figure 1—figure supplement 2d*).

In summary, genetic and epigenetic perturbations from steady state can cause a diversity of changes in HRAs that constrain the possible regulatory architectures consistent with experimental data obtained by perturbing them. HRAs that are nearly indistinguishable by genetic perturbation can be distinguished using epigenetic perturbations, underscoring the complementary nature of genetic and epigenetic perturbations.

## Changes in HRAs caused by single mutations form a sparse matrix

Just as mutations in a DNA genome can persist through replication at each cell division, changes in HRAs can persist by the formation of new positive feedback loops or the liberation of previously inhibited positive feedback loops. Six types of changes in sequence can arise from the four bases in a DNA genome upon mutation (A↔T, A↔G, A↔C, T↔G, T↔C, G↔C, with density of the change matrix = 1 [6/6]). To determine the analogous types of changes in regulatory architectures, the capacity for each of the 26 simplest HRAs (A to Z in *Figure 1*) to change into other HRAs was considered (*Figure 3*). Single perturbations of any interactor (entity/sensor) could result in the loss of the interactor, loss of an interaction, or a change in the polarity of an interaction (e.g. *Merdanovic et al., 2020*). These perturbations could 'mutate' the HRA by either collapsing the entire architecture or stably changing

it into a new HRA (*Figure 3*). Only changes that do not eliminate all positive regulatory inputs to an interactor can result in the persistence of a regulatory architecture rather than the eventual loss of one or more entities. Furthermore, since at steady state all gain of an entity/sensor is balanced by loss (via dilution at every cell division and/or other turnover mechanisms), any *permanent* reduction in the promotion of an entity/sensor will ultimately lead to its loss. Finally, if there is promotion of one sensor in a positive feedback loop and inhibition of another sensor in the same positive feedback loop then the net input can be positive or negative depending on the relative magnitudes of the inputs. With these considerations, enumeration of the changed HRAs that can result from a perturbation revealed that the 26 HRAs can be mutated to generate 61 different changes (24 through loss of interaction alone, 21 through change in polarity of interaction alone, and 16 through either change in regulation or though loss of an entity, with density of the change matrix ≈0.19 [61/325]). Thus, unlike changes in DNA sequence, not all changes are immediately accessible among HRAs (change matrix of 1 vs 0.19, respectively). Nevertheless, the heritable information transmitted using regulatory architectures is vast because even two or three interactors can form 26 heritable architectures that are collectively capable of 61 changes through single perturbations. This capacity is an underestimate because, single mutations can also result in the gain of new interactions that combine multiple HRAs into larger regulatory architectures with more interactors.

The constrained transition from one HRA at steady state to the next adjacent HRA through a single change (*Figure 3*) could skew the frequencies of different HRAs observed in nature and restrict the mechanisms available for development and/or evolution. For example, the HRA 'A' is accessible from 16 other HRAs but 'C' and 'T' are each accessible only from one other HRA ('J' and 'U', respectively). Furthermore, HRAs that rely on all components for their production ('C', 'F', 'H', 'K', and 'T') cannot change into any other HRAs from steady state without the addition of more positive regulation because any permanent loss in regulation without compensatory changes in turnover will result in the ultimate collapse of the entire architecture. These constraints can be overcome if change can occur through regulatory architectures that are not indefinitely heritable.

Deducing regulatory architectures from outcomes after perturbations is complicated by multiple HRAs resulting in the same HRA when perturbed (e.g. 16 HRAs can result in 'A' when perturbed, *Figure 3*). While measurement of dynamics after perturbations of *each* entity/sensor in turn can distinguish between all 26 architectures, the temporal resolution required is not obvious. This difficulty in accurate inference is apparent even for the simplest of perturbation experiments when inference relies only on end-point measurements, which are the most common in experimental biology. For example, a common experimental result is the loss of one regulator ($x$, say) leading to an increase in another ($y$, say), which is frequently interpreted to mean $x$ inhibits $y$ (*Figure 1—figure supplement 2e*). However, an alternative interpretation can be that $y$ promotes $x$ and itself via $z$, which competes with $x$ (*Figure 1—figure supplement 2e*). In this scenario, removal of $x$ leads to relatively more promotion of $z$, which leads to a relative increase in $y$. These equivalent outcomes upon loss or reduction of an entity in different architectures highlight the difficulty of inferring the underlying regulation after perturbation of processes with feedback loops. Therefore, simulations that enable exploration of outcomes when different interactors are perturbed at different times could enhance the understanding of underlying complexity, reduce biased inference, and better guide the next experiment.

## Simple Entity-Sensor-Property systems enable exploration of regulatory architectures

The most commonly considered regulatory networks (*Barabási and Oltvai, 2004*) are either limited in scope and/or are not causal in nature. For example, gene regulatory networks (*Levine and Davidson, 2005*) consider transcription factors, promoter elements, and the proteins made as the key entities. Additional specialized networks include protein-protein interaction networks (e.g. *Li et al., 2017*), genetic interaction networks (e.g. *Costanzo et al., 2019*), and signaling networks (e.g. *Azeloglu and Iyengar, 2015*). However, regulation of any process can rely on changes in a variety of molecules within cells, ranging from small molecules such as steroid hormones to organelles such as mitochondria. Furthermore, experimental studies often seek to provide explanations of phenomena in terms of a diversity of interacting entities. A common expression for all possible regulatory networks that preserves both causation and heritability can be derived by parsing all the contents of the bottleneck stage between two generations (e.g. one-cell zygote in the nematode *C. elegans*) into entities, their

sensors, and the sensed properties (*Jose, 2020b*). An additional advantage of such entity-sensor-property systems is that it is possible to consider entities that are sensed by a particular sensor even via unknown intermediate steps, allowing for simulation of regulatory networks despite incomplete knowledge of regulators.

For a given genome sequence, the number of distinguishable configurations of regulators represented as entities and sensors is given by the following equation (*Jose, 2020b*):

$$c_{tot} = \sum_{i=1}^{b} e_i \left( \sum_{j=1}^{s_i} s_j \left( \sum_{k=1}^{p_j} p_k \right) \right) \tag{1}$$

where, $e$ is the measured entity (total $b$ in the bottleneck stage between generations: $n_b$ in system, $o_b$ in environment), $s$ is the measuring sensor (total $s_i$ for $i^{th}$ entity), which is itself a configuration of entities drawn from the total $N$ per life cycle (i.e., $f(Y)$, with each $Y \subseteq \{e_1, e_2, ..., e_N\}$), and $p$ is the attainable and measurable values of the property measured by the sensor (total $p_j$ for the $j^{th}$ sensor of the $i^{th}$ entity). An entity or configuration of entities is considered a sensor only if changes in its values can change the values of other entities in the system or in the environment at some later time.

The complex summation $\sum_i e_i \sum_j s_j \sum_k p_k$ can be simplified into a product of measurable property values of individual entities/sensors if the possible numbers of property values of every entity/sensor combination is independent:

$$c_{tot} = \prod_{i=1}^{b} \prod_{j=1}^{s_i} n_{ij} \tag{2}$$

where, $n_{ij} = |\{p_1, p_2, \ldots p_j\}|$ is the number of property values as measured by the $j^{th}$ sensor of the $i^{th}$ entity. However, this upper limit is never reached in any system because regulatory interactions make multiple sensors/entities covary (*Jose, 2020b*). As a simple example, consider two sensors that activate each other in a Boolean network with no delay: the only possible values are {0,0} when both sensors are off and {1,1} when both sensors are on, because the mutual positive regulation precludes {0,1} and {1,0}.

Three simplifying assumptions were made to facilitate the simulation of regulatory networks with different architectures: (1) let the number of molecules be the only property measured by all sensors, (2) let each sensor be a single kind of molecule, and (3) let all entities be within the system. In these simple Entity-Sensor-Property (ESP) systems, the number of possible configurations is given by:

$$c_{tot} = \sum_{i=1}^{E} e_i \left( \sum_{j=1}^{s_i} n_j \right) < \prod_{i=1}^{E} \prod_{j=1}^{s_i} n_{ij} \tag{3}$$

where, $e$ is the measured entity (total $E$ in the system), $s$ is the measuring sensor (total $s_i$ for $i^{th}$ entity), $n_j$ is the attainable and measurable numbers of the $i^{th}$ entity, and $n_{ij}$ is the attainable and measurable numbers of the $i^{th}$ entity as measured by the $j^{th}$ sensor. In such simplified ESP systems, a sensor is simply an entity in the system that responds to changes in numbers of an entity by changing the numbers of that entity or another entity. Whether downstream changes occur depends on the sensitivity of the sensor and the step-size of changes in property value (i.e. number) of the downstream entity/sensor.

The Entities-Sensors-Property (ESP) framework is not entirely equivalent to a network, which is much less constrained. Nodes of a network can be parsed arbitrarily and the relationship between them can also be arbitrary. In contrast, entities and sensors are molecules or collections of molecules that are constrained such that the sensors respond to changes in particular properties of other entities and/or sensors. When considered as digraphs, sensors can be seen as vertices with positive indegree and outdegree. The ESP framework can be applied across any scale of organization in living systems and this specific way of parsing interactions also discretizes all changes in the values of any property of any entity (*Figure 1—figure supplement 2f*). In short, ESP systems are networks, but not all networks are ESP systems. Therefore, the results of network theory that remain applicable for ESP systems need further investigation.

To simulate ESP systems (e.g. *Figure 4a*) a hybrid approach with deterministic and stochastic aspects was used (see Methods for details). Specifically, the systems represented by *equation (3)* were

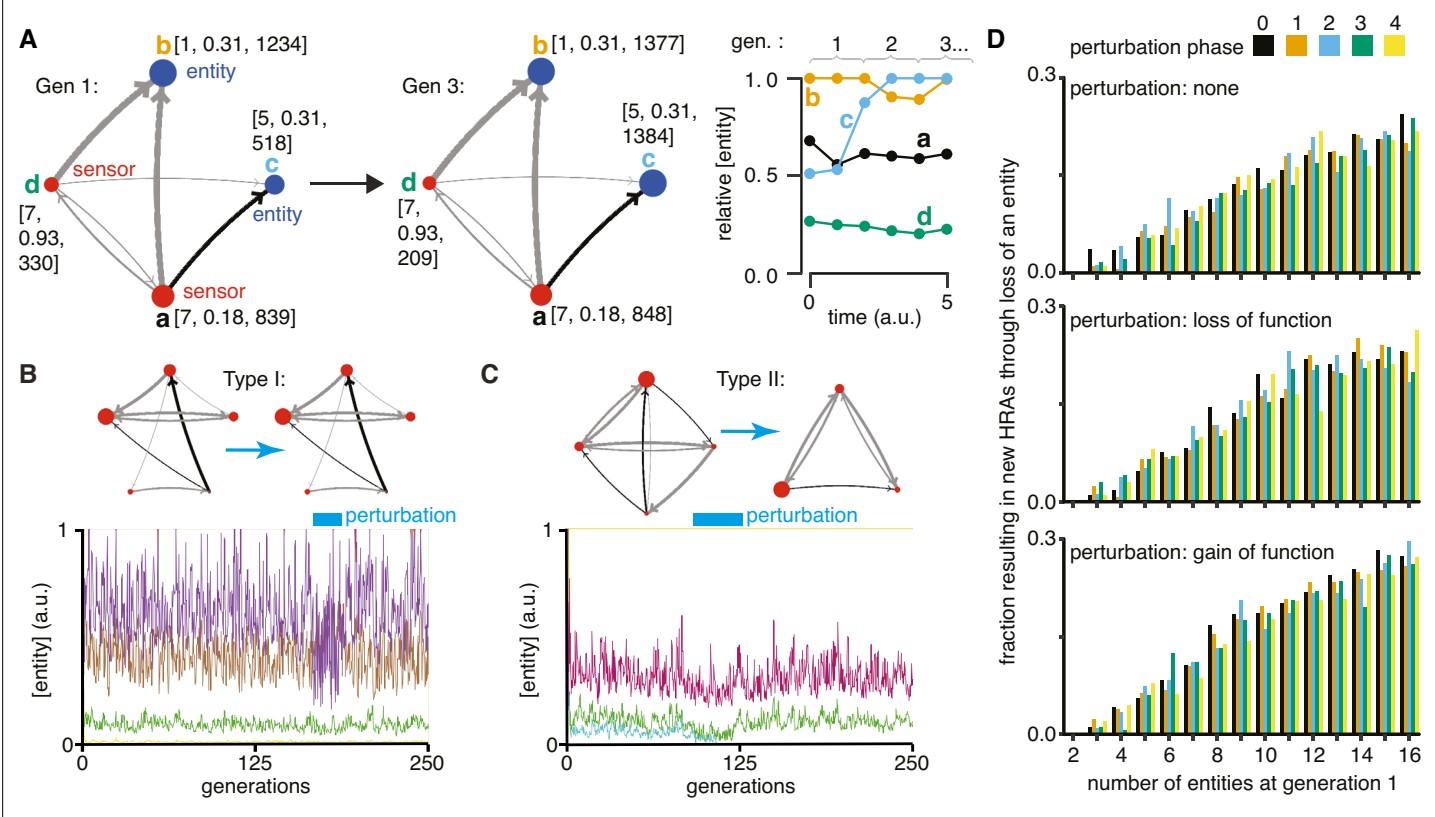

**Figure 4.** Regulatory architectures can be simulated as entity-sensor-property systems to examine how they persist or change in response to transient perturbations. (**A**) An ESP system illustrating the stability of a regulatory architecture despite changes in the relative numbers of the interactors (entities/sensors) over time. *Left*, Simulation of an ESP system showing how interacting molecules create regulatory architectures. This system consists of four entities (**a, b, c, d**), where 'd' and 'a' are also sensors. Each sensor (red) sends regulatory input (grey, positive or black, negative) to increase or decrease another sensor or entity (blue). Numbers of each entity (i.e., its property value) change in fixed steps per unit time. The number of sensors needed to cause one unit of change in property differs for each regulatory input (lower number = thicker line, representing lower threshold for downstream change). Each entity is depicted with property step, active fraction, and number at the start of the first generation (gen 1) and at the end of the third generation (gen 3). *Right*, The relative numbers of the entities, which can be together considered as 'phenotype', can change over time. Note that relative amounts of 'a', 'b', or 'd' remain fairly constant, but that of 'c' changes over time. (**B and C**) ESP systems can differ in their response to epigenetic change. *Top*, ESP systems are depicted as in A. *Bottom*, Relative abundance of each entity/sensor (different colors) or 'phenotype' across generations. Blue bars = times of epigenetic perturbation (reduction by two fold). In response to epigenetic perturbation that lasts for a few generations, Type I systems recover without complete loss of any entity/sensor (**B**) and Type II systems change through loss of an entity/sensor (**C**). (**D**) ESP systems of varying complexity can show heritable epigenetic changes, depending on when the system is perturbed. The numbers of randomly chosen entities were unperturbed (none, *top*), reduced to half the minimum (loss of function), or increased to twice the maximum (gain of function, *bottom*) every 50 generations for 2.5 generations and the number of systems responding with a new stable regulatory architecture that lasts for >25 generations were determined. Perturbations were introduced at each of five different time points with respect to the starting generation (phase - 0,1,2,3,4). Of the 78,285 stable systems, 14,180 showed heritable epigenetic change.

The online version of this article includes the following figure supplement(s) for figure 4:

**Figure supplement 1.** Key features of the ESP system explorer.

**Figure supplement 2.** Example of a system with long but finite stability.

**Figure supplement 3.** Characteristics of randomly sampled HRAs simulated with partitioning of entities during each cell division or generation and periodic perturbations.

simulated with random values for the numbers of each entity/sensor, the sensitivity of each sensor (i.e. the change in number needed to change a downstream entity/sensor), the step-size of changes in property (i.e. numbers changed per input from a sensor), and the active fraction (i.e. proportion that are available for interactions at any time). Only an arbitrary fraction of each entity (fixed over time) was simulated as active to account for processes such as folding, localization, diffusion, etc., that can limit regulatory interactions. Simulations were begun with a total of 500 molecules. A maximal increase of

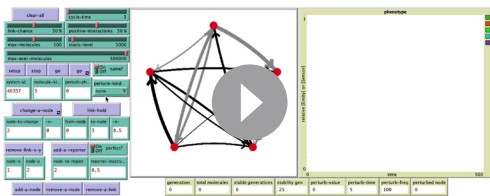

**Video 1.** NetLogo run showing the single-system explorer with sample interactions with the simulation. https://elifesciences.org/articles/92093/figures#video1

**Video 2.** Example ESP system with system-id 46357 without any perturbation. https://elifesciences.org/articles/92093/figures#video2

5000 molecules was allowed per cell cycle before each cell division to account for depletion of precursors, reactants, or building blocks (which were not explicitly simulated). The simulation was ended if total number of simulated molecules increased beyond a maximal number (500,000 in *Figure 4*) to account for the limited capacity of a cell. Thus, with each time step, the numbers of all entities/sensors changed deterministically based upon the randomly established initial regulatory architecture with stochastic changes in the numbers of each entity arising from the random order of evaluation at each time step and the reduction by ~1/2 at the start of each cell cycle, which simulates experimentally observed noise (e.g. *Wan et al., 2021*). With these parameters, regulatory architectures were simulated and the relative concentration of each entity/sensor was plotted over time (*Figure 4a*). These profiles represent the change in 'phenotype' over time and are akin to measurements of relative RNA abundance using RNA-seq (*Van den Berge et al., 2019*) or relative protein abundance using proteomic approaches (*Mund et al., 2022*). Thus, ESP systems represent networks or graphs with weighted edges (because of the different activities of different sensors needed for the transmission of each regulatory change), complex nodes (because of the multiple properties of each entity/sensor), and transmission delays (because of the variation in the duration of each regulatory interaction). Notably, the relative concentrations of an entity can change over time (e.g. 'c' in *Figure 4a*) while the underlying regulatory architecture is preserved (see *Figure 4—figure supplement 1*, *Video 1*, and run 'ESP_systems_single_system_explorer_v1.nlogo' in NetLogo [*Wilensky, 1999*] for exploration).

## An interactive simulation can be used to explore simple ESP systems

To gain intuitions by exploring and perturbing simulated ESP systems, several interactive features were added to the ESP simulator (*Figure 4—figure supplement 1*, *Video 1*). These include parameters that control the setup and running of randomly generated ESP systems by specifying the probability of regulatory interactions in the system (link-chance, *Figure 4—figure supplement 1*), the probability of positive versus negative interactions (positive-interactions, *Figure 4—figure supplement 1*), the maximum number of molecules at the start of the simulation (max-molecules, *Figure 4—figure supplement 1*), the maximum number of molecules that will arrest growth until dilution through cell divisions to simulate depletion of raw materials or energy (stasis-level, *Figure 4—figure supplement 1*), maximum number of molecules in total reflecting the limited space occupied by living systems (max-ever-molecules, *Figure 4—figure supplement 1*), and duration of the cell cycle (cycle-time, *Figure 4—figure supplement 1*). Particular systems can be re-established and re-simulated by setting the random number seed that is used for controlling all stochastic steps (system-id, *Figure 4—figure supplement 1*) and by additionally specifying the above parameters along with the number of entities/sensors at the start of the simulation (molecule-kinds, *Figure 4—figure supplement 1*). For each such system, the number of entities that can increase or decrease at one time was set to be characteristic of each entity/sensor (unit change in property value, i.e. number) and the number of sensors needed to change one unit of each entity/sensor was set to be characteristic of each regulatory interaction (thickness of link increases with increasing sensitivity of regulation). Periodic loss-of-function or gain-of-function perturbations (perturb-kind, *Figure 4—figure supplement 1*) can be set up to begin in five different phases relative to the start of the simulation (perturb-phase [0, 1, 2, 3 or 4], *Figure 4—figure supplement 1*). Perturbations that can be made during the simulation include changing the number of molecules of any entity/sensor (change-a-node, *Figure 4—figure supplement 1*), adding an entity/sensor (add-a-node, *Figure 4—figure supplement 1*), removing an entity/sensor (remove-a-node, *Figure 4—figure supplement 1*), removing a particular regulatory interaction (remove-link-x-y, *Figure 4—figure supplement 1*), removing a random regulatory interaction

(remove-a-link, *Figure 4—figure supplement 1*), and changing the strength of a regulatory input (link-hold, *Figure 4—figure supplement 1*). Finally, a reporter for any entity/sensor (add-a-reporter, *Figure 4—figure supplement 1*) can be set up that either perfectly or partially interacts with all its regulators (perfect?, *Figure 4—figure supplement 1*). A perfect reporter of an entity/sensor receives the same regulatory input as the entity/sensor of interest does. An imperfect reporter of an entity receives input from the same sensors as the entity/sensor of interest, but the polarity and strength of the input can vary. Regulatory outputs of the entity/sensor are not recreated for any reporter. As these changes are being made, both the regulatory architecture (Fig. *Figure 4—figure supplement 1*, *top*), which is re-drawn if the levels of any entity/sensor reaches zero, and the 'phenotype' as captured by the profile of relative concentrations of entities/sensors (*Figure 4—figure supplement 1*, *bottom*) can be observed.

## ESP systems differ in their susceptibility to heritable epigenetic changes

Explorations of ESP systems revealed that some systems can be stable for a large number of cell divisions (or equivalently generations) before the level of an entity/sensor becomes zero (e.g. system-id 62795 was stable for 59,882.5 generations (*Figure 4—figure supplement 2*)). This long, yet finite duration of stability highlights the difficulty in claiming any architecture is heritable forever if one of its entities/sensors has low abundance and can be lost with a small probability. Systems responded to transient perturbations that reduce the levels of a randomly chosen entity/sensor in two major ways: Type I systems recovered relative levels of entities/sensors and maintained the same regulatory architecture; Type II systems changed by losing one or more entities, resulting in new relative levels of entities/sensors, and new regulatory architectures that persisted for many subsequent generations. These two types were observed even when the numbers of some entities/sensors were changed by just twofold for a few generations (*Figure 4b* versus *Figure 4c*).

For systematic analysis, architectures that could persist for ~50 generations without even a transient loss of any entity/sensor were considered HRAs. Each HRA was perturbed (loss-of-function or gain-of-function) after five different time intervals since the start of the simulation (i.e. phases). The response of each HRA to such perturbations were compared with that of the unperturbed HRA. For loss-of-function, the numbers of one randomly selected entity/sensor were held at half the minimal number of all entities/sensors for 2.5 generations every 50 generations. This perturbation is like the loss of transcripts through RNA interference. For gain-of-function, the numbers of one randomly selected entity/sensor were held at twice the maximal number of all entities/sensors for 2.5 generations every 50 generations. This perturbation is like overexpression of a particular mRNA or protein. Of 225,000 ESP systems thus simulated, 78,285 had heritable regulatory architectures that remained after 250 generations (system-ids and other details for exploration of individual systems are in *Supplementary file 2*). These persistent systems included entities/sensors with relative numbers that changed over time as well as those with nearly constant relative numbers. For each number of interactors considered (2–16), only a fraction of the regulatory architectures were stable, plateauing at ~50% of simulated systems with 10 or more entities/sensors (*Figure 4—figure supplement 3*, top left). This plateau is likely owing to the limit set for the maximal number of molecules in the system because most systems with many positive regulatory links quickly reached this limit. Although systems began with up to 16 entities/sensors, by the end of 250 generations, there was a maximum of ~8 entities/sensors and a median of ~4 entities/sensors in stable systems (*Figure 4—figure supplement 3*, *bottom left*). Furthermore, an excess of positive regulatory links was needed to sustain a system with negative

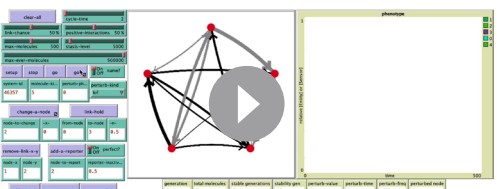

**Video 3.** Example ESP system with system-id 46357 and with loss-of-function perturbations in phase 0.
https://elifesciences.org/articles/92093/figures#video3

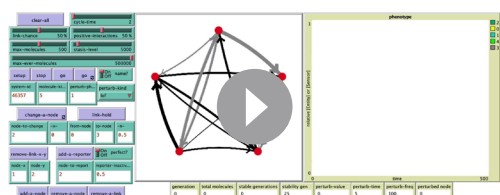

**Video 4.** Example ESP system with system-id 46357 and with loss-of-function perturbations in phase 1.
https://elifesciences.org/articles/92093/figures#video4

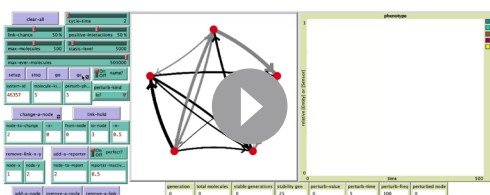

**Video 5.** Example ESP system with system-id 46357 and with loss-of-function perturbations in phase 2.
https://elifesciences.org/articles/92093/figures#video5

**Video 6.** Example ESP system with system-id 46357 and with loss-of-function perturbations in phase 3.
https://elifesciences.org/articles/92093/figures#video6

regulatory links (*Figure 4—figure supplement 3*, right), with the minimal system that can sustain a negative regulatory interaction requiring three sensors, as expected. This bias reflects the inability of negative regulatory interactions alone to maintain regulatory architectures over time across cell divisions (*Figure 1—figure supplement 2g*).

## Periodic rescue or perturbation can expand the variety of heritable regulatory architectures

Since the relative abundance of each entity/sensor is expected to trace a transgenerational waveform (*Jose, 2020c*) and to fluctuate with each time step, the relative timing of the perturbations (i.e. their phase) can impact the persistence of particular regulatory architectures. Specifically, entities/sensors with low numbers could be rescued from reaching zero by well-timed gain-of-function perturbations and those with high numbers could be rescued from arresting the simulation by well-timed loss-of-function perturbations. Therefore, the fractions of persistent ESP systems that showed heritable epigenetic change through loss of one or more entities were examined by starting with 2–16 molecules for each phase and type of perturbation (*Figure 4d*). As expected, only HRAs with a minimum of three entities/sensors at the start of the simulation showed heritable epigenetic changes. Fractions of HRAs showing heritable epigenetic changes were comparable across all perturbations for a given number of starting entities/sensors, with more such HRAs identified with increasing numbers of starting entities/sensors (compare *Figure 4d*, top, middle, and bottom). The variations in the numbers identified for different phases of perturbation when starting with a particular number of entities/sensors was comparable to the variation observed in systems that were not perturbed (compare *Figure 4d*, top with *Figure 4d*, middle and bottom), suggesting that no particular phase is more effective. Although some architectures were unaffected by all perturbations, many showed an altered response based on both the nature and phase of the perturbations. Consider the behavior of the illustrative example defined by system-id 46357 that begins with five entities/sensors (see *Videos 2–9*, *Video 10*, *Video 11*, *Video 12*). The unperturbed ESP system stabilizes with an architecture of three entities (of the type 'E' in *Figure 1*) until ~74 generations, when one of the entities is lost and the new architecture (of the type 'A' in *Figure 1*) remains stable until ~286.5 generations. However, perturbations yield a variety of different stabilities depending on phase and type of perturbation. Periodic loss-of-function perturbations with phase '0', '3', or '4' resulted in stability of all 3 entities until collapse at ~130.5, ~181.5, or ~99.5 generations, respectively. But, such perturbations with phase '1' resulted in an earlier change from type 'E' to type 'A' with collapse at ~193.5 generations and with phase '2' resulted in a later change from 'E' to 'A' with collapse at ~184.5 generations. On the other hand, periodic gain-of-function perturbations with phase '0', '1', '2', or '4' prolonged the type 'E' architecture

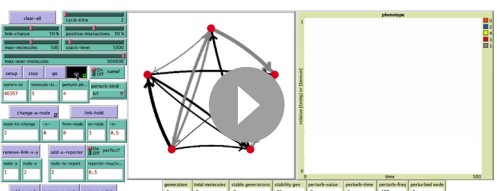

**Video 7.** Example ESP system with system-id 46357 and with loss-of-function perturbations in phase 4.
https://elifesciences.org/articles/92093/figures#video7

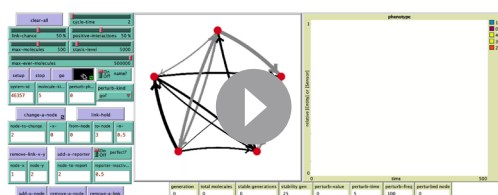

**Video 8.** Example ESP system with system-id 46357 and with gain-of-function perturbations in phase 0.
https://elifesciences.org/articles/92093/figures#video8

until ~306, ~253, ~212, or ~708 generations, respectively, after which the new type 'A' architecture persisted beyond 1000 generations, by when the simulation was ended. However, such perturbations with phase '3' preserved the type 'E' architecture until collapse at ~311.5 generations.

Collectively, these results reveal that the heritability of regulatory architectures that are intrinsically unstable can be enhanced through interactions that alter the regulation of one or more sensors. Such external regulators could be part of the environment (e.g. periodic interactions such as circadian signals) or other cells (e.g. periodic interactions with somatic cells for regulatory architectures transmitted along a germline).

## Organismal development can permit HRAs that incorporate interactions with somatic cells and intergenerational delays in regulation

While for unicellular organisms, each cell division results in a new generation, for multicellular organisms, transmission of heritable information across generations occurs along a lineage of cells that can include many cell divisions with periods of quiescence and interactions with somatic cells. The nematode *C. elegans* is a well-characterized multicellular organism (*Corsi, 2006*) that has many features such as the early separation of the germline during development, sexual reproduction, and the generation of different somatic tissues (*Figure 5a*, top), that can be incorporated into simulations and are useful for generalization to other animals, including humans. For example, 14 cell divisions are necessary to go from the zygote of one generation to the zygote of the next (*Supplementary file 3*). The loading of oocytes with maternal molecules in multiple organisms makes one generation of delay in regulation (i.e. maternal regulation) common, but such ancestral effects could last longer in principle (e.g. grandparental effects are easily imagined in humans because oocytes begin developing within the female fetus of a pregnant woman). Indeed, studies on RNA silencing in *C. elegans* have revealed long delays in regulation within the germline. For example, parental *rde-4* can enable RNA silencing in adult *rde-4*(-) progeny (*Marré et al., 2016*). Furthermore, loss of *meg-3/–4* can result in persistent RNA silencing defects despite restoration of wild-type *meg-3/–4* for multiple generations (*Dodson and Kennedy, 2019*; *Lev et al., 2019*; *Ouyang et al., 2019*). However, examining the impact of such long delays and of interactions with somatic cells (e.g. *Abdu et al., 2016*) is computationally expensive. Therefore, to simulate regulatory architectures that persist from one zygote to the next while satisfying some of the known constraints of *C. elegans* lineage and development, the ESP simulator was modified to incorporate the observed timings of cell division versus growth along the germline (*Supplementary file 3*, based on *Oegema and Hyman, 2006*; *Bao et al., 2008*; *Giurumescu et al., 2012*; *Kimble and Crittenden, 2005*; *Jaramillo-Lambert et al., 2007*; *Hirsh et al., 1976*) and allow a delay of up to two generations for the impact of a regulatory interaction (*Figure 5b*).

Since many genes expressed in the germline can be required for fertility or viability, the analysis of how they are regulated across generations poses a challenge. Following the behavior of a reporter across generations provides a proxy that can be used to understand transgenerational regulation in a relatively wild-type background (i.e. the tracer approach [*Jose, 2018*]). However, regulatory interactions that rely on the gene product will not be re-created by the reporter. For example, the mRNA sequence used to produce antisense small RNA of the gene will differ from that used for the reporter.

To simulate the regulation of a germline gene and its reporter, a modified ESP system is required. In addition to a minimal positive feedback loop required for heritability, positive and negative regulators that depend on *cis*-regulatory sequences shared by both the gene and its reporter as well as

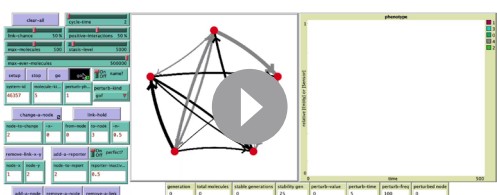

**Video 9.** Example ESP system with system-id 46357 and with gain-of-function perturbations in phase 1.
https://elifesciences.org/articles/92093/figures#video9

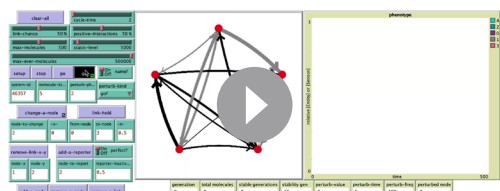

**Video 10.** Example ESP system with system-id 46357 and with gain-of-function perturbations in phase 2.
https://elifesciences.org/articles/92093/figures#video10

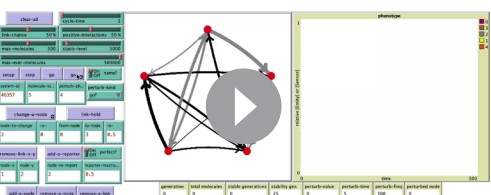

**Video 11.** Example ESP system with system-id 46357 and with gain-of-function perturbations in phase 3.
https://elifesciences.org/articles/92093/figures#video11

**Video 12.** Example ESP system with system-id 46357 and with gain-of-function perturbations in phase 4.
https://elifesciences.org/articles/92093/figures#video12

such regulators that depend on the different products (e.g. the mRNA and/or protein of gene versus reporter) need to be simulated. Of the 10,000 such ESP systems simulated, 11 maintained all entities/ sensors for 10 generations (*Supplementary file 4*). A representative such system (*Figure 5c*) forms a HRA that incorporates regulatory delays ranging from 0 to 171 hr and can persist for hundreds of generations. Examining the levels of all entities/sensors over the first 10 generations (*Figure 5d*) reveals that despite starting at random values the HRA settles with a reproducible pattern during each generation (gen 2 onwards in *Figure 5d*). The transgenerational waveforms traced by the relative numbers of all entities/sensors reveal periods of increased expression/activity for some entities/ sensors during development (red asterisks in *Figure 5d*), as observed for many genes expressed in the germline. Future studies that obtain data on key regulators with spatial and temporal resolution can be used to discriminate between different HRAs that drive the expression of different genes of interest.

## Tuning of positive feedback loops acting across generations can explain the dynamics of heritable RNA silencing in *C. elegans*

The simple fact that organisms resemble their parents in most respects provides evidence for the homeostatic preservation of form and function across generations. Yet, this 'transgenerational homeostasis' (*Jose, 2018*) is overcome in some cases such that epigenetic changes persist for many generations (reviewed in *Fitz-James and Cavalli, 2022*). Indeed, DNA methylation patterns of unknown origin are thought to have persisted for millions of years in the fungus *C. neoformans* (*Catania et al., 2020*). Studies on RNA silencing in *C. elegans* (reviewed in *Frolows and Ashe, 2021*) provide strong evidence for heritable epigenetic changes in the expression of particular genes, facilitating analysis. When a gene expressed in the germline is silenced using double-stranded RNA of matching sequence and/or germline small RNAs called piRNAs, the silencing can last from one or two generations to hundreds of generations (*Devanapally et al., 2021*; *Shukla et al., 2021*; *Priyadarshini et al., 2022*). The maintenance of RNA silencing across generations is thought to require a positive feedback loop formed by antisense small RNAs called 22G RNAs that are bound to the Argonaute HRDE-1 (*Buckley et al., 2012*) and sense mRNA fragments processed into poly-UG RNAs (pUG RNAs) (*Shukla et al., 2020*) that can act as templates for RNA-dependent RNA polymerases that synthesize the 22G RNAs. However, this mechanism does not explain the variety of effects that can arise when such genes with long-term silencing are exposed to other genes of matching sequence (*Seth et al., 2018*; *Devanapally et al., 2021*). For example, when a gene silenced by disrupting RNA regulation within the germline (*iT*, a transgene with silenced *mCherry* sequences) is exposed to genes with matching sequences (*Figure 6*, *Devanapally et al., 2021*), different outcomes are possible. After initial silencing in trans, the newly exposed genes can recover from silencing (*mCherry* and *mCherryΔpi* in *Figure 6a*) when separated from the source of silencing signals, but can be either continually silenced (*mCherry/iT* in *Figure 6a*) or become resistant to silencing (*mCherryΔpi/iT* in *Figure 6a*) depending on the presence of intact piRNA-binding sequences despite the continued presence of the source. While this observation suggests that recognition of the target mRNA by piRNAs prolongs RNA silencing, loss of the Argonaute PRG-1 that binds piRNAs and regulates more than 3000 genes in the germline (e.g. *Reed et al., 2020*) also prolongs the duration of heritable RNA silencing (*Shukla et al., 2021*). Although the release of shared regulators upon loss of piRNA-mediated regulation in animals lacking PRG-1 could be adequate to explain enhanced HRDE-1-dependent transgenerational silencing initiated by dsRNA in *prg-1(-)* animals, such a competition model alone cannot explain the observed alternatives of susceptibility, recovery and resistance (*Figure 6a*). Recent considerations of such competition for

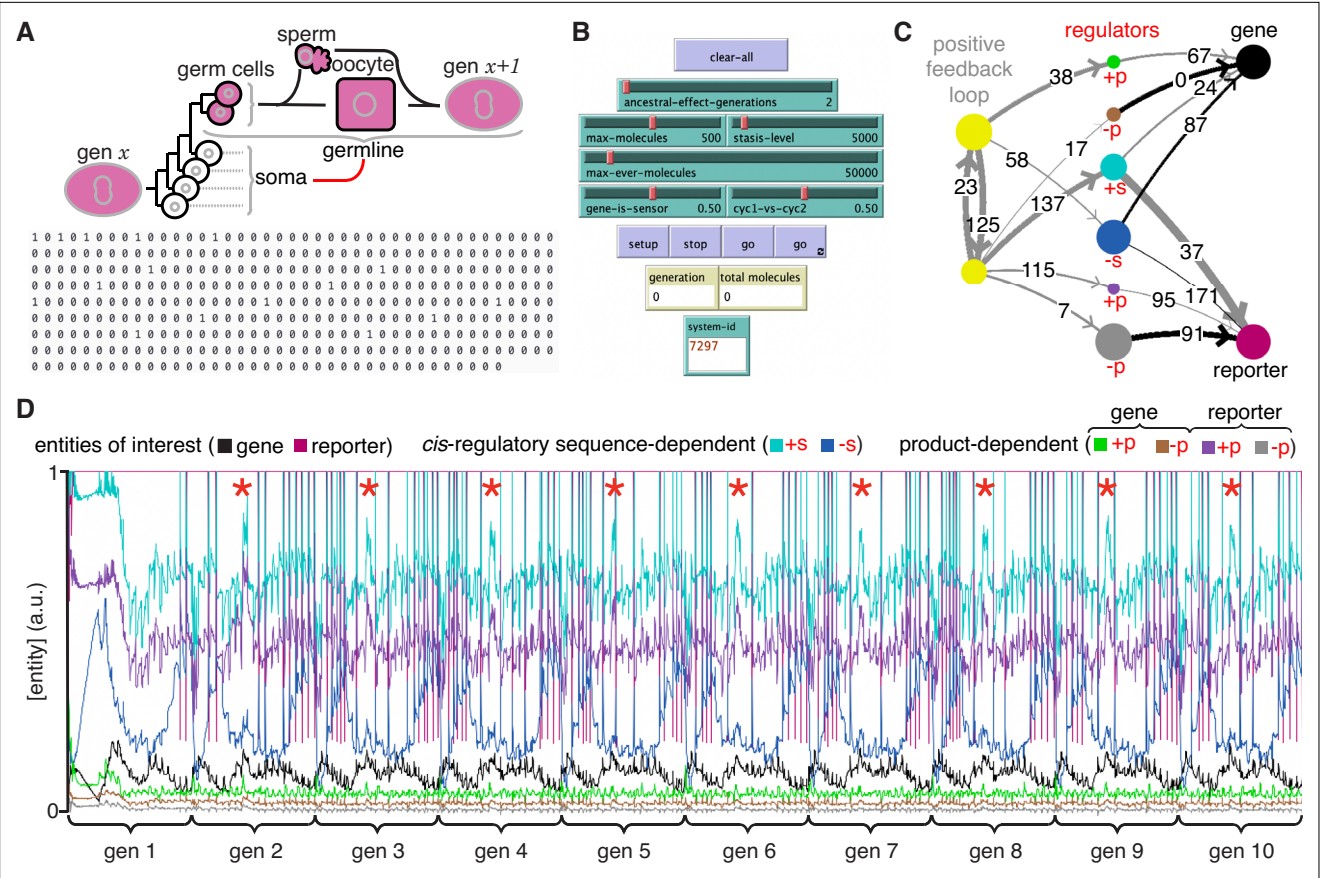

**Figure 5.** ESP systems that incorporate the timings of cell division during *C. elegans* development and temporal delays in regulatory interactions can recreate periods of increased expression in every generation. (**A**) *Top*, Schematic of cell divisions between two successive generations of *C. elegans*. Cells that maintain the intergenerational continuity through cell divisions (magenta, germline), cells that cannot contribute to the next generation through cell divisions (white, soma) but arise in each generation (gen *x* and gen *x+1*) from the bottleneck stage, and the interactions between these two cell types (red line) are depicted. *Bottom*, Experimentally determined timing of cell division (1) versus growth (0) from one zygote to the next in *C. elegans* in 15 min intervals (=1 time step in simulations), which give a generation time of ~91.25 hr (=365 time steps). See *Supplementary file 3* for the relative timing of cell divisions based on past studies. (**B**) Key control features for simulating HRAs that incorporate organismal timing of cell divisions and temporal delays in regulation. In addition to controls used in the single system explorer (*Figure 4—figure supplement 1*), the following sliders were added: one to set the number of generations of ancestors that can contribute regulation (ancestral-effect-generations, e.g., 2 for parental effects), one to set the probability of the regulatory origin for each interaction from one of the two sensors that form the positive feedback loop required for heritability (cyc1-vs-cyc2), and one to set the probability of the gene of interest being a sensor providing regulatory input into the positive feedback loop instead of an entity (gene-is-sensor). Monitors that show the current generation and the total number of molecules, and an input to set the system-id were also added. (**C**) Representative simulated HRA that incorporates temporal delays and the characteristic timings of cell divisions in *C. elegans*. Different types of positive (+) and negative (-) regulators (red) that depend on *cis*-regulatory sequences (+s and -s, e.g. transcription factors), and that depend on the gene product (+p and -p for gene and reporter, for example small RNAs made using mRNA template, chaperones that promote the folding of the protein, etc.) are depicted with color coded arrows (+, grey and -, black). Different relative delays in regulation (hours on arrows, maximum of 2 x generation time to allow for the widely observed parental regulation) are also depicted. The unknown components of the core positive feedback loops required for heredity were simulated as two sensors that promote each other's production in addition to the production of all other entities/sensors. (**D**) Relative concentrations of entities/sensors regulated by the HRA in (**C**) over 10 generations showing transgenerational waveforms. Properties, active fractions, relative numbers, and regulatory interactions were considered and relative numbers of each entity/sensor depicted as in *Figure 4* with colors as in (**C**). Although the simulation began with random numbers for all entities/sensors, the HRA settles into a reproducible pattern within two generations with periods of increased relative concentrations for some entities/sensors in every generation (red asterisks). Also see *Video 13*.

regulatory resources in populations of genes that are being silenced suggest explanations for some observations on RNA silencing in *C. elegans* (*Karin et al., 2023*). Specifically, based on Little's law of queueing, with a pool of M genes silenced for an average duration of T, new silenced genes arise at a rate $\lambda$ that is given by M = $\lambda$ T. However, this theory cannot predict which gene is silenced at any given time, why some genes are initially susceptible to silencing but subsequently become resistant

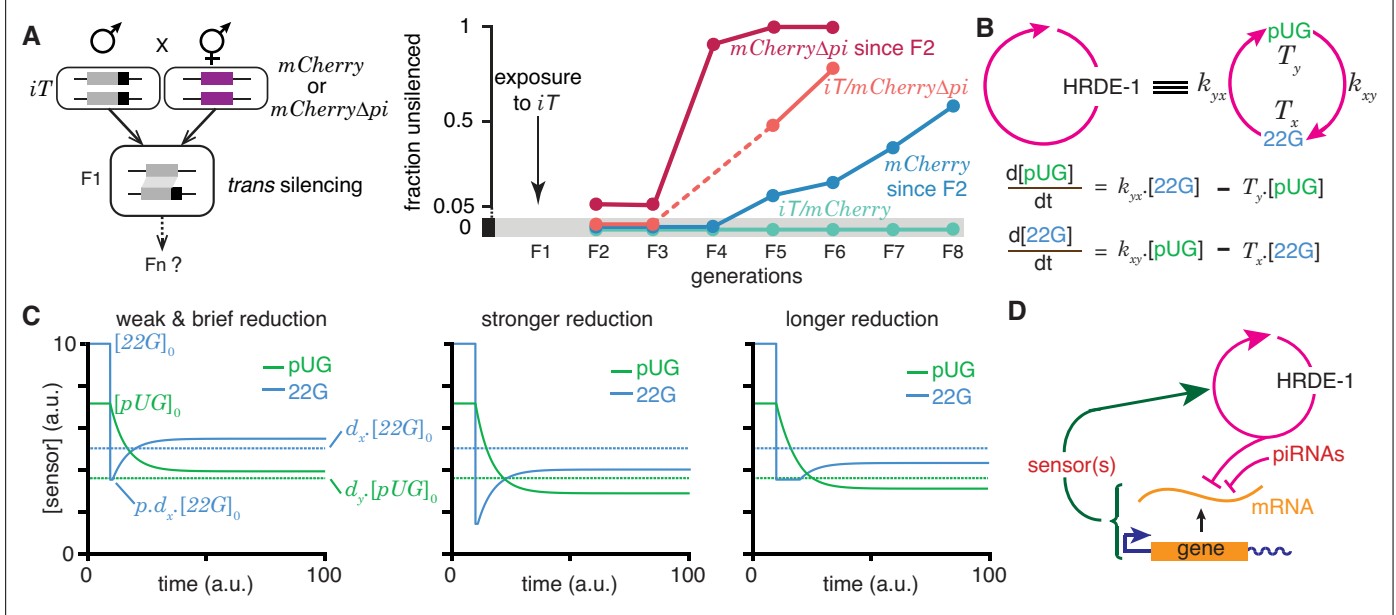

**Figure 6.** Regulation of a positive feedback loop can explain the magnitude and duration of experimentally observed heritable RNA silencing. (**A**) Experimental evidence from *C. elegans* for susceptibility to, recovery from, and resistance to *trans* silencing by a silenced gene(**A**) has been adapted from Figure 5A of *Devanapally et al., 2021*). *Left*, Schematic of experiment showing a gene silenced for hundreds of generations by mating-induced silencing (iT = *mex-5p::mCherry::h2b::tbb-2 3' utr::gpd-2 operon::gfp::h2b::cye-1 3' utr*) exposed to genes with matching sequences (*mCherry* and *mCherryΔpi*, i.e. *mCherry* without piRNA binding sites) to initiate *trans* silencing. *Right*, Dynamics of heritable RNA silencing showing the initial exposure to *trans* silencing by iT (F1 generation), subsequent recovery after separation from iT ('*mCherry* since F2' and '*mCherryΔpi* since F2'), resistance to silencing by iT (iT/mCherryΔpi), or persistence of silencing by iT (iT/mCherry). Fractions of animals that recover *mCherry* or *mCherryΔpi* expression (fraction unsilenced) are depicted with error bars eliminated for simplicity. (**B**) Abstraction of the HRDE-1-dependent positive feedback loop required for the persistence of RNA silencing. *Top*, Representation of the mutual production of RNA intermediates (22G and pUG) with rates of production ($k_{yx}$ and $k_{xy}$) and turnover ($T_x$ and $T_y$). *Bottom*, Ordinary differential equations for the rates of change of pUG RNAs (pUG) and 22G RNAs (22G). See text for details. (**C**) Impact of transient epigenetic perturbations on subsequent activity of a positive feedback loop. *Left*, response to a brief and weak reduction in the levels of one sensor (22G) of the positive feedback loop. The steady-state levels after recovery were above the threshold required for a silencing effect (dotted lines). Steady states ($[22G]_0$ and $[pUG]_0$), perturbation level ($p.d_x. [22G]_0$), and levels required for silencing ($d_x. [22G]_0$ and $d_y. [pUG]_0$) are indicated. *Middle* and *Right*, Stronger (*middle*) or longer (*right*) reduction can result in steady-state levels after recovery being below the threshold required for a silencing effect (dotted lines). (**D**) Deduced regulatory architecture that explains data shown in (**A**) by including enhancement of silencing by piRNA binding on target mRNA and a gene-specific inhibitory loop that can act across generations through as yet unidentified sensor(s). Prolonged silencing in *prg-1(-)* animals (*Shukla et al., 2021*) suggests that these sensor(s) are among the genes mis-regulated in *prg-1(-)* animals (e.g. *Reed et al., 2020*). See *Figure 6—figure supplement 1* for depictions of additional equivalent architectures.

The online version of this article includes the following figure supplement(s) for figure 6:

**Figure supplement 1.** Equivalent representations of a transgenerational feedback loop that can tune HRDE-1-dependent heritable RNA silencing.

(e.g. *mCherryΔpi* in *Figure 6a*), and why the silencing of some genes can last for hundreds of generations. Thus, there is a need for understanding the origins of gene-specific differences in the dynamics of heritable epigenetic changes.

22G RNAs and pUG RNAs are experimentally measurable molecular markers whose levels are thought to be proportional to the extent of gene silencing (e.g. *Gu et al., 2009*; *Pak et al., 2012*; *Shirayama et al., 2012*), although formally gene-specific regulatory features could influence this proportionality (*Knudsen et al., 2023*). To understand how the activity of the underlying positive feedback loop that maintains the levels of these RNAs could relate to the extent of observed silencing, the HRDE-1-dependent loop was abstracted into a minimal positive feedback loop with 22G RNAs and pUG RNAs promoting each other's production (*Figure 6b*, *top*). Ordinary differential equations (*Figure 6b*, *bottom*) were developed for interdependent change in both RNAs by considering their rates of promotion ($k_{xy}$ for 22G and $k_{yx}$ for pUG) and turnover ($T_x$ for 22G and $T_y$ for pUG). All molecules and chemical modifications that are necessary to maintain a positive feedback loop are sensors because they need to transmit the change from an 'upstream' regulator to a 'downstream' regulator. This 22G-pUG-positive feedback loop thus represents mutual promotion by two sensors (i.e.

the HRA 'A' in *Figure 1*). When any changed molecule or chemical modification is thus viewed as one component of a heritable regulatory architecture driven by a positive feedback loop, two criteria that impact the duration of epigenetic changes through the reduction of particular sensors are immediately suggested: (1) for every permanent change that is observed, all sensors that participate in the regulatory loop must be reduced to a level below that required for observing the change; and (2) for eventual recovery from a change, at least one sensor should be above the threshold required to drive the increase of all other sensors above their respective levels for observing the change. Consistently, a weak and brief reduction of 22G RNAs from steady state levels results in the eventual recovery of both 22G RNA and pUG RNA levels above the threshold required for them to be effective for silencing (*Figure 6c*, *left*). Stronger (*Figure 6c*, *middle*) or longer (*Figure 6c*, *right*) reductions can result in new steady-state levels for both RNAs that are below the threshold required for silencing.

To obtain an analytic expression for how long heritable RNg needs to be inhibited for eventual recovery, the impact of transient reduction in the activity of the $22G\text{-}pUG$ positive feedback loop from steady state ($[22G]_0$ and $[pUG]_0$) was considered. Let $d_x.[22G]_0$ or lower be insufficient for silencing and let 22G RNAs be transiently perturbed to $p.d_x.[22G]_0 \neq 0$, where $d_x < 1$ and $p < 1$. The critical duration ($t_{22G}$) of such a perturbation for permanent reduction of 22G RNA below the level required for silencing is given by

$$t_{22G} > \frac{1}{T_y} ln \left[ \frac{\frac{1}{d_x.p} - 1}{\left(\frac{1}{p} - 1\right)\left(\frac{T_y}{T_x} + 1\right)} \right]$$

(4)

where $T_x$ and $T_y$ are the rates of turnover for 22G RNAs and pUG RNAs, respectively. Analogously, the critical duration ($t_{pUG}$) of such a perturbation for permanent reduction of pUG RNA below the level required for silencing is given by

$$t_{pUG} > \frac{1}{T_x} ln \left[ \frac{\frac{1}{d_y.p} - 1}{\left(\frac{1}{p} - 1\right)\left(\frac{T_x}{T_y} + 1\right)} \right]$$

(5)

where $d_y.[pUG]_0$ or lower is insufficient for silencing. Derivation of the general case for these equations (HRA 'A') is presented in Methods. These equations suggest that depending on the parameters of the architecture, different sensors may be more easily perturbed to cause heritable epigenetic changes. For example, for the same critical threshold below steady state ($d_x = d_y = 0.5$) and the same extent of perturbation ($p = 0.8$), an architecture with $[22G]_0$ = 10, $[pUG]_0$ = 7.14, $T_x$ = 0.05, $T_y$ = 0.1, $k_{xy}$ = 0.07, $k_{yx}$ = 0.0714, is more quickly inhibited by reducing 22G RNAs than by reducing pUG RNAs (6.93 vs 27.72 units of time).

Combining these considerations with the observations that both piRNA binding to target mRNAs (*Figure 6a*) and loss of the piRNA-binding Argonaute PRG-1 (*Shukla et al., 2021*; *Priyadarshini et al., 2022*) prolong the duration of heritable RNA silencing suggests a unified mechanism that sets gene-specific durations of heritable RNA silencing. Specifically, the dynamics of recovery from silencing depends on the strength of an inhibitory feedback that can act across generations to reduce the HRDE-1-dependent positive feedback loop. This transgenerational inhibition relies on sensor(s) that are regulated by PRG-1 and is opposed by piRNA binding to the silenced mRNA (*Figure 6d*).

This proposed mechanism implies the existence of sensor(s) that respond to the activity of the HRDE-1-dependent positive feedback loop by recognizing one or more of the molecules and/or chemical modifications generated. Consistently, a chromodomain protein HERI-1 has been reported to be recruited to genes undergoing heritable RNA silencing and is required to limit the duration of the silencing (*Perales et al., 2018*). Additional sensors are likely among the >3000 genes mis-regulated in animals lacking PRG-1 (*Reed et al., 2020*; *Shukla et al., 2021*). The levels or activities of these sensor(s) could either increase or decrease in response to the activity of the HRDE-1-dependent loop depending on which of the multiple equivalent configurations of the negative feedback are present at different genes (expected to decrease in *Figure 6—figure supplement 1*, *left* and *right*, but increase in *Figure 6—figure supplement 1*, *middle*). However, in every case, the net result is a reduction in the activity of the HRDE-1-dependent loop (*Figure 6—figure supplement 1*). Therefore, genes encoding

such sensors could be among those that show increased mRNA levels (e.g. 2517 genes in *Reed et al., 2020*) and/or that show decreased mRNA levels (e.g., 968 genes in *Reed et al., 2020*) upon loss of PRG-1. Another set of genes that could encode similar sensors are those identified using repeated RNAi as modifiers of transgenerational epigenetic kinetics (*Houri-Ze'evi et al., 2016*). Regardless of the identities of such sensors, differences in the transgenerational feedback that reduces some component(s) of the 22G-pUG positive feedback loop can explain the persistence of, recovery from, and resistance to heritable RNA silencing when different genes are targeted for silencing.

In summary, experimental evidence and theoretical considerations suggest that the HRDE-1-dependent positive feedback loop that generates 22G RNAs and pUG RNAs is tuned by negative feedback that acts across generations to cause different durations of heritable RNA silencing. Such tuning can explain silencing for a few generations followed by recovery from silencing as well as resistance to silencing. Future studies are required for testing the quantitative predictions on the impact of reducing 22G RNA or pUG RNA levels (*Equations (4) and (5)*) and for identifying the PRG-1-dependent genes that could have roles in the transgenerational inhibition of heritable RNA silencing (*Figure 6d*).

## Discussion

The framework presented here establishes criteria for (1) heritable information in regulatory architectures, (2) the persistence of epigenetic changes across generations, (3) distinguishing between regulatory architectures using transient perturbations, (4) making unstable regulatory architectures in the germline heritable through interactions with somatic cells, and (5) generating epigenetic changes of defined magnitude and duration.

### ESP systems can be used to analyze many types of regulatory interactions

The simple ESP systems simulated here (*Figure 4*) can be extended to include a wide variety of properties to explore heritable epigenetic changes that can occur in cell/organelle geometry, phase separation, protein folding, etc. In general, each kind of molecule or entity can have multiple properties that are sensed by different sensors. For example, concentration, folded structure, primary sequence, and subcellular localization of a protein could each be measured by different sensors that respond by causing different downstream effects. If a protein ($x$) is regulated by three different regulators that each change its concentration ($C$), subcellular localization ($L$), or folded structure ($F$), then the protein $x$ could be considered as having $C$, $L$, and $F$ as values for its regulated properties. If the protein $x$ in turn acts as a regulator that changes another entity $y$, then the activity of $x$ regulating $y$ could be simulated as a combined function of its concentration, localization and folded structure (i.e. activity of $x=f(C, L, F)$). Such simulations preserve both the independent regulation of different properties of a protein along with the potential equivalence in the activity of a higher concentration of partially folded proteins and a lower concentration of well-folded proteins. Thus, appropriate mapping onto an ESP system would enable the explanation of many phenomena in terms of regulatory architectures formed by any set of interactors while rigorously considering heritability.

### A positive feedback loop can only support the inheritance of one property of an entity

While no entity can promote changes in all its properties by itself, some entities can promote changes in one of their properties through self-regulatory interactions under some conditions. For example, prions can act as replicating stores of information that template changes in the conformation of other proteins with the same sequence (*Scheckel and Aguzzi, 2018*), although other properties of prions such as their concentration, subcellular localization, rate of turnover, etc. are determined through interactions with other entities/sensors. Such self-regulatory interactions for the control of some properties can be considered by allowing self-referential loops. Specifically, if the protein $x$ above were a prion, then its properties will include $C$, $L$, and $F$ as above, with the value of $F$ changing with time as a function of both concentration and prior proportion folded (i.e. $F(t+1)=g(C, F(t))$). Similar considerations underscore that any one positive feedback loop can promote only one property of an entity and not all of its properties. For example, consider small RNAs that are associated with gene silencing

in *C. elegans*. The targeting of a gene by small RNAs could be preserved in every generation through a regulatory loop whereby recognition of mRNA by antisense small RNAs results in the production of additional small RNAs by RNA-dependent RNA Polymerases. However, the mere existence of this feedback loop cannot explain the different concentrations of small RNAs targeting different genes (*Gu et al., 2009*) or the different durations of persistent small RNA production when initiated experimentally (*Devanapally et al., 2021*). Similar considerations apply for chromatin modifications, DNA modifications, RNA modifications, and all other 'epigenetic marks'.

## Mutability of epigenetic information changes non-monotonically with complexity

Inducing heritable changes in epigenetic information is more challenging than inducing similar changes in genetic information (*Jose, 2020c*). Chemically altering a single molecule (typically DNA) is sufficient for inducing a genetic change, however, similarly altering one entity of a regulatory architecture (say, a protein) to induce an epigenetic change requires simultaneously altering the many copies of that entity without altering DNA sequence. All DNA bases with induced chemical changes are deleted or converted into one of the other bases by the replication and repair pathways. Consequently, only 6 different base exchanges are possible in DNA sequence through a single mutation, but even when only up to three interactors are considered, 61 different HRA changes are possible through a single mutation (*Figure 3*). Importantly, after a heritable change, the DNA sequence remains similarly mutable, but the impact of a change (genetic or epigenetic) on subsequent mutability of a regulatory architecture could increase or decrease. Consider a change that incorporates a new transcription factor into a regulatory architecture. If it is an activator, it could promote the expression of many genes leading to the incorporation of more RNAs and proteins into the regulatory architecture. Conversely, if it is a repressor, it could repress the expression of many genes leading to the removal of RNAs and proteins from the regulatory architecture. Either of these consequences could make the entire architecture more robust such that drastic perturbation is needed to cause observable change. When such robust architectures occur during development, the identity of a cell could become relatively fixed and heritable through cell divisions and yet remain compatible with specific natural (*Jarriault et al., 2008*) and/or induced (*Shi et al., 2017*) cell fate transformations. Thus, such cell fate determination reflects the acquisition of different robust states, providing the appearance of a cell 'rolling down an epigenetic landscape' (*Waddington, 1957*).

## Complexity of heritable regulatory architectures

Despite imposing heritability, regulated non-isomorphic directed graphs soon become much more numerous than unregulated non-isomorphic directed graphs as the number of interactors increase (125 vs 5604 for 4 interactors, *Table 1*). With just 10 interactors, there are $>3 \times 10^{20}$ unregulated non-isomorphic directed graphs (*Sloane, 2021*) and HRAs are expected to be more numerous. This tremendous variety highlights the vast amount of information that a complex regulatory architecture *can* represent and the large number of changes that are possible despite sparsity of the change matrix (*Figure 3*). This number is potentially a measure of epigenetic evolvability – the ability to adapt and survive through regulatory change without genetic mutations. However, architectures made up of numerous interactors that are all necessary for the positive feedback loop(s) required for transmitting information across generations are more vulnerable to collapse (e.g., 'C' and 'T' in *Figure 1*). Furthermore, spatial constraints of the bottleneck stage (one cell in most cases) present a challenge for the robust transmission of regulatory information. A speculative possibility is that complex architectures are compressed into multiple smaller positive feedback loops for transmission between generations with the larger HRAs being re-established through interactions between the positive feedback loops in every generation. Examples of such numerous but small positive feedback loops include small RNA-mediated, chromatin-mediated, and prion-mediated loops that specify the identity of genes for particular forms of regulation in the next generation. However, determining how the rest of the regulatory information is transmitted in each case and whether such compression with redundancy is a general principle of heredity require further study.

## Information density in living systems

Individual entities transmitted across generations can be considered as carrying part of the heritable information if such information is always seen in the context of the sensors that interact with the entity. While the maximal information that can be carried by DNA sequence is a product of its length ($l$) and the number of bits contributed by each base ($2l = \log_2[4].l$), the maximal number of relevant bits contributed by any entity - including the genome - depends on the number of states sensed by all interacting sensors (for $i^{th}$ entity = $\log_2(\sum_{j=1}^{s_i} s_j \left( \sum_{k=1}^{p_j} p_k \right))$ from *Equation 1*; *Jose, 2020b*). The genome likely interacts with the largest number of sensors per molecule, making it the entity with the most information density. When a gene sequence is transcribed and translated, sequence information is transmitted from one molecule (DNA) to many (RNA(s) and/or protein(s)), thereby reducing the density of sequence information per entity (e.g. RNA or protein of a particular sequence). However, information is also added to these entities through the process of RNA folding and protein folding, which depend on interactions with the surrounding chemical and physical context (e.g. *Porter and Looger, 2018*). The resultant structural (and potentially catalytic) specialization of RNAs and proteins provides them with additional properties that are relevant for other sensors, thereby increasing their information content beyond that in the corresponding genome sequence. Importantly, these proteins and RNAs can interact to create molecular complexes and organelles that again concentrate information in higher-order entities present as fewer copies within cells (e.g. centrosomes). Such complexes and organelles therefore are entities with high information density. In this view, the information density throughout a bottleneck stage connecting two generations (e.g. the single-cell zygote) is non-uniform and ranges from entities with very high density (e.g. the genome) to those with very low density (e.g. water). Beginning with the simplest of regulatory architectures (*Figure 1*), progressive acquisition of entities and their interacting sensors would lead to the incorporation of regulators with increasing information density. An entity that is connected to numerous sensors that each respond to changes in a fraction of its properties can appear to be the chief 'information carrier' of the living system (e.g. DNA in cells with a genome). Thus, as heritable regulatory architectures evolve and become more complex, entities with a large number of sensed properties can appear central or controlling (see *Dyson, 1982*; *Noble, 2008* for similar ideas). This pathway for increasing complexity through interactions since before the origin of life suggests that when making synthetic life, any form of high-density information storage that interacts with heritable regulatory architectures can act as the 'genome' analogous to DNA.

# Methods

## Key resources table

| Reagent type (species) or resource | Designation | Source or reference | Identifiers | Additional information |
|---|---|---|---|---|
| Software, algorithm | Python | https://www.python.org/downloads/release/python-385/ | | |
| Software, algorithm | R | https://cran.r-project.org/bin/macosx/ | | |
| Software, algorithm | NetLogo | https://ccl.northwestern.edu/netlogo/ | | |
| Software, algorithm | Gephi | https://gephi.org/ | | |

## Software

All calculations were performed by hand or using custom programs in Python (v. 3.8.5), and/or R (v. 3.6.3). Simulations and analyses were performed using NetLogo (v. 6.1.1), Python (v. 3.8.5), and/or R (v. 3.6.3). Transitions between HRAs were depicted using the circular layout plugin in Gephi (v. 0.10.1 202301172018). All programs used in this study are available at https://github.com/AntonyJose-Lab/Jose_2023, copy archived at *Jose, 2023*.

## Analysis of heritable regulatory architectures: i. Overview

The possible weakly connected non-isomorphic graphs that form regulatory architectures capable of indefinite persistence and the maximal information that can be stored using them ($\log_2 N$) were

calculated. Systems of ordinary differential equations (ODEs) were used to describe the rate of change for each interacting entity ($x$, $y$, $z$) in each of the 26 heritable regulatory architectures (A to Z) with the relative amounts of all entities at any time defined as the 'phenotype' at that time. Each regulatory architecture is characterized by a maximum of 9 parameters in addition to the relative amounts of each entity: a rate of turnover for each entity (3 total; $T_x$, $T_y$, $T_z$) and rates for each regulatory interaction between entities (6 total; e.g. $k_{xy}$ is the regulatory input to $x$ from $y$, $k_{yz}$ is the regulatory input to $y$ from $z$, etc.). Relative amounts of each interacting entity for each architecture at steady state ($x_0$, $y_0$, $z_0$) were determined by setting all rate equations to zero, which results in constraints on the variables for each architecture (e.g. for A: $y_0 = x_0 \cdot \frac{T_x}{k_{xy}} = x_0 \cdot \frac{k_{yx}}{T_y}$; for B: $y_0 = x_0 \cdot \frac{T_x}{k_{xy}} = x_0 \cdot \frac{k_{yx}}{T_y}$; $z_0 = x_0 \cdot \frac{T_x}{k_{xy}} \cdot \frac{k_{zy}}{T_z}$, etc.). These constraints were used to obtain a total of 128,015 parameter sets ($T_x$, $T_y$, $T_z$, $k_{xy}$, $k_{yz}$, etc.) that are compatible with steady state for each architecture (*Supplementary file 1*). Particular parameter sets were then used to illustrate the consequences of genetic and epigenetic changes in all architectures (*Figure 2* and *Figure 2—figure supplements 1–7*). The simpler expressions for steady-state levels for all regulatory architectures when there are no turnover mechanisms for any entities and the only 'turnover' occurs by dilution upon cell division ($T = T_x = T_y = T_z$) were derived. Expressions for steady state values when all entities are lost at a constant rate to complex formation ($\gamma$) and for changes upon complete loss of an entity were also derived.

The impacts of transient perturbations from steady state were examined for each architecture to identify conditions for heritable epigenetic change by defining thresholds ($d_x$, $d_y$, $d_z$) for observing a defect for each entity (e.g. for the function of $x$, $d_x \cdot x_0$ is not sufficient, when $d_x < 1$, or $d_x \cdot x_0$ is in excess, when $d_x > 1$). Responses after perturbations are illustrated for all architectures (*Figure 2—figure supplement 1*) and cases where genetic and epigenetic perturbations result in distinct responses are highlighted (*Figure 2*). The duration and extent of perturbation needed for heritable epigenetic changes were explored using numerical solutions of the equations describing the dynamics of each regulatory architecture. Explorations of 22G-pUG positive feedback loop were similarly performed by simulating a type 'A' HRA using ODEs (*Figure 6*). Analytical expressions for conditions that enable heritable epigenetic change were derived for the simplest heritable regulatory architecture where two entities ($x$ and $y$) mutually promote each other's production. Transitions between HRAs (*Figure 3*) were worked out manually by considering the consequence of each change from steady state for each HRA.

## Analysis of simple heritable regulatory architectures: ii. Details – derivations of equations

The dynamics of the 26 heritable regulatory architecture (HRAs in *Figure 1*) can be described by systems of ordinary differential equations. The rate of change of each entity ($x$, $y$, or $z$ for each HRA in *Figure 1*) can be written by aggregating positive and negative contributions from the other sensors (e.g. $k_{xy} \cdot y - k_{xz} \cdot z$ if $x$ is positively regulated by $y$ and negatively regulated by $z$) with loss of each entity described with a turnover term (e.g. $-T_x \cdot x$ for $x$). To ensure the concentrations of all entities remain non-negative, as expected in real living systems, the equations are bounded to be applicable only when the values of the changing entity is greater than zero. These equations can then be used to derive other equations and inequalities of interest.

### Steady-state relationships

At steady state, each architecture results in relative amounts of each entity ($x_0$, $y_0$, $z_0$) that could define a 'phenotype'. These relative concentrations can be derived by setting the rate of change of all entities to zero.

For the heritable regulatory architecture A,

$$\frac{dx}{dt} = k_{xy} \cdot y - T_x \cdot x, \ \forall \ (x + dx) > 0, \text{ else } (x + dx) = 0$$

$$\frac{dy}{dt} = k_{yx} \cdot x - T_y \cdot y, \quad \forall \ (y + dy) > 0, \text{ else } (y + dy) = 0$$

where, $k_{xy}$ is the rate constant for the production of $x$ promoted by $y$; $k_{yx}$ is the rate constant for the production of $y$ promoted by $x$; $T_x$ is the rate of turnover of $x$; and $T_y$ is the rate of turnover of $y$.

i.e., $\dot{X} = A.X$, where $\dot{X} = \begin{bmatrix} \dot{x} \\ \dot{y} \end{bmatrix}$; $A = \begin{bmatrix} -T_x & k_{xy} \\ k_{yx} & -T_y \end{bmatrix}$; $X = \begin{bmatrix} x \\ y \end{bmatrix}$

At steady state,

$$-T_x.x + k_{xy}.y = 0$$
$$k_{yx}.x - T_y.y = 0$$

that is, $A.X = 0$, where $A = \begin{bmatrix} -T_x & k_{xy} \\ k_{yx} & -T_y \end{bmatrix}$; $X = \begin{bmatrix} x_0 \\ y_0 \end{bmatrix}$, which has solutions that satisfy:

$$\frac{x_0}{y_0} = \frac{k_{xy}}{T_x} = \frac{T_y}{k_{yx}}$$

Equations for steady state and the resulting solutions for the other HRAs can be similarly derived (see previous version of the paper for details).

## Steady states with loss of all entities to complex formation at a constant rate

A common way in which entities change in living systems is through the formation of intermolecular complexes that then interact with different entities to perform different functions. If the same number of molecules per unit time ($\gamma$) are lost for all entities (e.g. through incorporation into a 1:1:1 stoichiometric complex), then each entity needs to grow at the same rate ($\gamma$) to maintain steady state ($x_0, y_0, z_0$).

For the heritable regulatory architecture A,

$$k_{xy}.y - T_x.x = \gamma$$
$$-T_y.y + k_{yx}.x = \gamma$$

that is $A.X = B$, where $A = \begin{bmatrix} -T_x & k_{xy} \\ k_{yx} & -T_y \end{bmatrix}$; $X = \begin{bmatrix} x_0 \\ y_0 \end{bmatrix}$; $B = \begin{bmatrix} \gamma \\ \gamma \end{bmatrix}$

The solution is given by $X = A^{-1}.B$

(For $A = \begin{bmatrix} a & b \\ c & d \end{bmatrix}$, $A^{-1} = \frac{1}{ad-bc} \begin{bmatrix} d & -b \\ -c & a \end{bmatrix}$)

$$X = \frac{1}{T_x.T_y - k_{xy}.k_{yx}} \begin{bmatrix} -T_y & -k_{xy} \\ -k_{yx} & -T_x \end{bmatrix} . \begin{bmatrix} \gamma \\ \gamma \end{bmatrix}$$

i.e., $\begin{bmatrix} x_0 \\ y_0 \end{bmatrix} = \frac{\gamma}{k_{xy}.k_{yx} - T_x.T_y} \begin{bmatrix} T_y + k_{xy} \\ T_x + k_{yx} \end{bmatrix}$

To similarly identify the rates of growth for the other heritable regulatory architectures (B to Z) under a constant rate of loss for all entities, inverses for the 3x3 matrices can be used (see previous version of the paper for details). If all molecules are diluted through cell division (typically one cell dividing to give two), then for maintaining steady state on average, each molecule needs to accumulate to twice the average steady-state value per cell cycle (See *Figure 4* for simulations that include cell divisions).

## Response to genetic loss of an entity

Loss of an entity (typically the RNA or protein product of a gene) through a genetic mutation is a common perturbation used for analyzing living systems.

For the heritable regulatory architecture A,

When $x$ is lost,

$$\dot{y} = -T_y.y$$

Which has the solution,

$y = y_0.e^{-T_y t}$, that is the concentration of $y$ undergoes exponential decay through turnover from its steady-state value ($y_0$).

Similarly, when $y$ is lost,

$$x = x_0.e^{-T_x t}$$

For all other architectures, loss of one entity can result in different dynamics of the other two entities ($\alpha$ and $\beta$, say) depending on their regulatory interactions. The equations for their dynamics is given by a pair of differential equations that can be coupled.

that is $\dot{X} = A.X$, where $\dot{X} = \begin{bmatrix} \dot{\alpha} \\ \dot{\beta} \end{bmatrix}$; $A = \begin{bmatrix} a & b \\ c & d \end{bmatrix}$; $X = \begin{bmatrix} \alpha \\ \beta \end{bmatrix}$.

The exact equations that result upon loss of each entity in each regulatory architecture can be derived for each HRA (see previous version of the paper for details) and potentially be used to distinguish the different architectures through genetic experiments (e.g. knockout of individual genes using genome editing). Since for the steady state of each architecture, there are a maximum of three equations, a maximum of three variables among rates of production ($k_{xy}$, $k_{yx}$ etc.), rates of turnover ($T_x$, $T_y$, $T_z$), and the steady-state concentrations ($x_0$, $y_0$, $z_0$) are constrained. Changes in all entities after each loss can be determined for all architectures through simulations by choosing random values for the unconstrained parameters (*Figure 2* and *Figure 2—figure supplements 1–7*, left).

## Response to epigenetic change

Analytic expressions for heritable epigenetic change after reducing the levels of a sensor from steady state are derived below for the simplest of heritable regulatory architectures 'A' (*Figure 1*).

The dynamics of two entities ($x$ and $y$) that mutually promote each other's production is given by a pair of differential equations that are coupled.

that is $\dot{X} = A.X$, where $\dot{X} = \begin{bmatrix} \dot{x} \\ \dot{y} \end{bmatrix}$; $A = \begin{bmatrix} a & b \\ c & d \end{bmatrix}$; $X = \begin{bmatrix} x \\ y \end{bmatrix}$ and $a = -T_x$, $b = k_{xy}$, $c = k_{yx}$, $d = -T_y$

The general solution for the concentrations $x(t)$ and $y(t)$ are:

$$x(t) = \frac{K_1}{2m} \left( e^{0.5t(a+d-m)} \left( e^{tm}(a-d+m) + m+d-a \right) \right) - \frac{bK_2}{m} \left( e^{0.5t(a+d-m)} - e^{0.5t(a+d+m)} \right)$$

$$y(t) = \frac{K_2}{2m} \left( e^{0.5t(a+d-m)} \left( e^{tm}(d-a+m) + m-d+a \right) \right) - \frac{cK_1}{m} \left( e^{0.5t(a+d-m)} - e^{0.5t(a+d+m)} \right)$$

where, $m = \sqrt{a^2 - 2ad + 4bc + d^2}$ and, $K_1$ and $K_2$ are constants.

Since the system was already at steady state before the perturbation, $T_x T_y = k_{xy} k_{yx}$,

$$m = \sqrt{T_x^2 - 2T_x T_y + 4k_{xy}k_{yx} + T_y^2} = T_x + T_y$$

Substituting the value of $m$ in the equations above and simplifying yields,

$$x(t) = \frac{K_1.T_y + K_2.k_{xy}}{T_x + T_y} + \frac{K_1.T_x - K_2.k_{xy}}{T_x + T_y}.e^{-(T_x+T_y)t}$$

and

$$y(t) = \frac{K_2.T_x + K_1.k_{yx}}{T_x + T_y} + \frac{K_2.T_y - K_1.k_{yx}}{T_x + T_y}.e^{-(T_x+T_y)t}$$

Let $d_x$ be the reduction in $x$ (reduction-of-function) needed to observe a defect when $x_0$ is the steady-state value before perturbation. That is, $d_x.x_0$ is not sufficient for the function of $x$ in a living system, where $d_x < 1$. Let $x$ be perturbed to $x_p = p.d_x.x_0 \neq 0$ from $t = 0$ until $t = t_p$, where $p < 1$. For heritable epigenetic changes using reduction-of-function perturbations ($d_x < 1$ and/or $d_y < 1$), which preserve the architecture at a new steady state: $x_{ps} < d_x.x_0$ and $y_{ps} < d_y.y_0$.

To determine the concentration of $y$ at the end of the perturbation ($y_p$) the equation $\dot{y} = x_p.k_{yx} - T_y.y$ can be solved using $y(t) = y_0$ at $t = 0$. The general solution of the equation is given by,

$$y(t) = \frac{x_p.k_{yx}}{T_y} + C_1.e^{-T_y t}$$

Substituting for $y(0) = y_0$ at $t = 0$, and rearranging gives $C_1 = \frac{y_0.T_y - x_p.k_{yx}}{T_y}$. Thus, at the end of the perturbation (i.e., at $t_p$),

$$y(t_p) = y_p = \frac{x_p.k_{yx}}{T_y} + \left(\frac{y_0.T_y - x_p.k_{yx}}{T_y}\right).e^{-T_y t_p}$$

The new steady states ($x_{ps}$ and $y_{ps}$) will be reached from the initial concentrations of $x_p$ and $y_p$. Therefore, to determine the new steady state, the initial values of $x_p$ and $y_p$ can be used at new $t = 0$ to get the values for the constants $K_1$ and $K_2$.

$$x_p = \frac{K_1.T_y + K_2.k_{xy}}{T_x + T_y} + \frac{K_1.T_x - K_2.k_{xy}}{T_x + T_y}.e^{-(T_x + T_y)t = 0}$$

$$y_p = \frac{K_2.T_x + K_1.k_{yx}}{T_x + T_y} + \frac{K_2.T_y - K_1.k_{yx}}{T_x + T_y}.e^{-(T_x + T_y)t = 0}$$

Which simplifies to,

$$x_p = \frac{K_1.T_y + K_2.k_{xy}}{T_x + T_y} + \frac{K_1.T_x - K_2.k_{xy}}{T_x + T_y}$$

$$y_p = \frac{K_2.T_x + K_1.k_{yx}}{T_x + T_y} + \frac{K_2.T_y - K_1.k_{yx}}{T_x + T_y}$$

Solving for each,

$$K_1 = x_p$$
$$K_2 = y_p$$

To obtain the new steady state value $x_{ps}$, set $t = \infty$ in the equation using the above constants.

$$x_{ps} = \frac{x_p.T_y + y_p.k_{xy}}{T_x + T_y}$$

$$y_{ps} = \frac{y_p.T_x + x_p.k_{yx}}{T_x + T_y}$$

Consider the equality that is the threshold for observing heritable epigenetic effects,

$$x_{ps} = \frac{x_p.T_y + y_p.k_{xy}}{T_x + T_y} = d_x.x_0$$

Substituting for $x_p$ and simplifying yields,

$$y_p.k_{xy} = d_x.x_0.T_x + d_x.x_0.T_y(1 - p)$$

Substituting for $y_p$

$$\left(\frac{x_p.k_{yx}}{T_y} + \left(\frac{y_0.T_y - x_p.k_{yx}}{T_y}\right).e^{-T_y t_p}\right).k_{xy} = d_x.x_0.T_x + d_x.x_0.T_y(1 - p)$$

Collecting exponential terms,

$$e^{-T_y t_p}\left(\frac{y_0.k_{xy}.T_y - p.d_x.x_0.k_{yx}.k_{xy}}{T_y}\right) = d_x.x_0.T_x + d_x.x_0.T_y(1 - p) - \frac{p.d_x.x_0.k_{yx}.k_{xy}}{T_y}$$

$$e^{-T_y t_p} = \frac{d_x.x_0.T_x.T_y + d_x.x_0.T_y.T_y.(1 - p) - p.d_x.x_0.k_{yx}.k_{xy}}{y_0.k_{xy}.T_y.T_y - p.d_x.x_0.k_{yx}.k_{xy}}$$

$$e^{-T_y t_p} = \frac{d_x.x_0.T_x.T_y + d_x.x_0.T_y.T_y - p.d_x.x_0.(k_{yx}.k_{xy} + T_y.T_y)}{y_0.k_{xy}.T_y - p.d_x.x_0.k_{yx}.k_{xy}}$$

Dividing numerator and denominator with $T_y$ ,

$$e^{-T_y t_p} = \frac{d_x.x_0.T_x + d_x.x_0.T_y - p.d_x.x_0.\left(\frac{k_{yx}.k_{yx}}{T_y} + T_y\right)}{y_0.k_{xy} - p.d_x.x_0.\frac{k_{yx}.k_{yx}}{T_y}}$$

At steady state, the ratio $\frac{x(t)}{y(t)}$ will be independent of the concentrations of $x$ and $y$. That is, $\frac{x_{ps}}{y_{ps}} = \frac{x_0}{y_0} = \frac{k_{xy}}{T_x} = \frac{T_y}{k_{yx}}$ . Therefore, these equalities can be used to simplify the above equations. Substituting $\frac{k_{yx}.k_{yx}}{T_y} = T_x$ ,

$$e^{-T_y t_p} = \frac{d_x.x_0.T_x + d_x.x_0.T_y - p.d_x.x_0.\left(T_x + T_y\right)}{y_0.k_{xy} - p.d_x.x_0.T_x}$$

Substituting , $y_0.k_{xy} = x_0.T_x$

$$e^{-T_y t_p} = \frac{d_x.x_0.T_x + d_x.x_0.T_y - p.d_x.x_0.\left(T_x + T_y\right)}{x_0.T_x - p.d_x.x_0.T_x}$$

Dividing numerator and denominator by $x_0.T_x$

$$e^{-T_y t_p} = \frac{d_x + d_x.\frac{T_y}{T_x} - p.d_x.\left(1 + \frac{T_y}{T_x}\right)}{1 - p.d_x}$$

Simplifying,

$$e^{-T_y t_p} = \frac{\left(1 + \frac{T_y}{T_x}\right).\left(1 - p\right).d_x}{1 - p.d_x}$$

Dividing numerator and denominator by $d_x$

$$e^{-T_y t_p} = \frac{\left(1 + \frac{T_y}{T_x}\right).\left(1 - p\right)}{\left(\frac{1}{d_x} - p\right)}$$

Taking the $log_e$ on both sides,

$$-T_y t_p = ln\left[\frac{\left(1 + \frac{T_y}{T_x}\right).\left(1 - p\right)}{\frac{1}{d_x} - p}\right]$$

that is

$$t_p = \frac{1}{T_y} ln\left[\frac{\frac{1}{d_x} - p}{\left(1 + \frac{T_y}{T_x}\right).\left(1 - p\right)}\right]$$

Dividing numerator and denominator within the antilogarithm by $p$,

$$t_p = \frac{1}{T_y} ln\left[\frac{\frac{1}{d_x.p} - 1}{\left(\frac{1}{p} - 1\right)\left(1 + \frac{T_y}{T_x}\right)}\right]$$

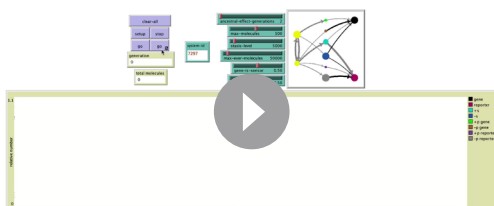

**Video 13.** NetLogo run showing an ESP system with regulatory delays and developmental timing of cell divisions adapted from experimental results in *C. elegans*.
https://elifesciences.org/articles/92093/figures#video13

This equation relates the duration of a perturbation ($t_p$) and the extent of the perturbation ($p < 1$ for loss-of-function) beyond the threshold that causes a defect in the function of $x$ (i.e. $d_x$). Increasing the duration of the perturbation beyond $t_p$ for a given extent of perturbation ($p$) will result in heritable epigenetic change where the steady-state levels of both interactors are insufficient for appropriate function.

Similarly, the minimal duration of perturbation for heritable epigenetic changes through a defect in the function of $y$ is given by,

$$t_p > \frac{1}{T_x} ln \left[ \frac{\frac{1}{d_y.p} - 1}{\left(\frac{1}{p} - 1\right)\left(1 + \frac{T_x}{T_y}\right)} \right]$$

These inequalities were verified using numerical simulations (see 'HRA_A_ tp_analytical_expression_check.py') and additional HRAs were similarly simulated to gain intuitions about the consequences of epigenetic reduction in the levels of entities (*Figure 2* and *Figure 2—figure supplements 1–7*).

## Analysis of entity-sensor-property systems: i. Overview

Simple ESP systems were simulated using custom programs in NetLogo (*Wilensky, 1999*)(details are available within the 'code' and 'info' tabs of each program). Results from systematic explorations obtained by running NetLogo programs from the command line (i.e. 'headless') were analyzed using R (e.g. *Figure 4d*, *Figure 4—figure supplement 3*). The developmental timings of cell divisions in *C. elegans* were curated manually from the literature (*Supplementary file 3*) and used to simulate ESP systems that incorporate developmental timings and temporal delays in regulation (*Figure 5*). Videos (*Videos 1–13*) were made by recording screen captures of NetLogo runs using QuickTime Player.

## Analysis of entity-sensor-property systems: ii. Details – simulation of simple ESP systems

A model was created in NetLogo to simulate entity-sensor-property systems and their evolution across generations for exploring regulatory architectures (ESP_systems _explorer_v1.nlogo). A variety of regulatory architectures were simulated using this model to identify ones with some architecture that persisted for 250 generations with or without perturbations (e.g. 78285 out of 225000 tested using the experiment 'ESP_origins_2–16_mols' described under behaviorspace). Each stable ESP system can be further analyzed in detail using the related ESP_systems_single_system_explorer_v1. All randomly chosen values for parameters in the regulatory architectures that lead to stability, that is heritable for many generations, can be recreated because the random-seed is set for each run using the behaviorspace-run-number. The behaviorspace-run-number serves as the 'system-id' in the related ESP_systems_single_system_explorer_v1.nlogo.

For each system, entities/sensors were defined as 'turtles' with 3 variables that stored its attributes:

1. val – a variable for storing an entity/sensor's current 'property value' (e.g. concentration, % conformational change, sequence, etc.). It changes throughout the simulation and is always represented in architectures as the size of the circle for each entity after scaling it relative to all extant entities.
2. property – a variable for storing the steps of change by which values (i.e. the 'val' above) can change if a positive or negative interaction crosses the threshold for change. For these simulations, it is characteristic of the entities/sensors themselves and does not change as the system evolves through interactions, which introduces the simplification that every sensor sees the same property of a given entity/sensor.
3. inactive-fraction – a variable for storing the fraction of entity/sensor not available for regulatory interactions at each time step (tick) because of processes like protein folding,

compartmentalization, diffusion, etc. This is a characteristic of each entity/sensor that is randomly chosen at the beginning of the simulation and does not change during the simulation.

The regulatory interactions in each system were specified using 'links' that were weighted to indicate the threshold required for the regulatory interaction and colored to indicate the nature of the regulation. Specifically, the links have two variables:

1. weight - a variable that indicates the number of sensors needed to change one unit of property for each entity. This parameter is characteristic of each regulatory interaction and captures the threshold needed for transmission of change. For display, the thickness of the regulatory link is set to be 0.5 - weight / 20. Thus, a lower threshold for transmission is represented as a wider link.
2. color - a variable that indicates whether the regulatory interaction is positive (grey) or negative (black).

Parameters that were varied in the exploration of ESP systems were:

1. molecule-kinds, which was the number of entities/sensors that are part of the regulatory architecture.
2. perturb-kind (none, lof, or gof), which was a chooser for perturbing a random entity/sensor every ~50 generations for 2.5 generations by increasing (gof) or decreasing (lof) its value (i.e. concentration/number) by two fold of the maximal or minimal values, respectively, of all the entities/sensors.
3. perturb-phase, which was the precise timing for starting the periodic perturbations (e.g. 0=starting @ tick 100; 1=starting @ tick 101; 2=starting @ tick 102; 3=starting @ tick 103; 4=starting @ tick 104)

Additional parameters, which were not varied in the exploration of ESP systems were:

1. cycle-time, which was set at 2 and represented the timing in ticks for each generation.
2. link-chance, which was set at 50% and gave the probability that any two entities/sensors will interact when the system is set up at the beginning of the simulation.
3. positive-interactions, which was set at 50% and gave the probability that a regulatory interaction is positive.
4. max-molecules, which was set at 500 and was the maximal number of total molecules at the start of the simulation.
5. stasis-level, which was set at 5000 and was the number of molecules that arrests growth until molecules get diluted upon cell division.
6. max-ever-molecules, which was set at 500000 was the maximal number of molecules of all kinds put together that can be within any system at any time. This limit simulates living systems existing in a finite environment.

Monitors reporting behaviorspace-run-number (system-id), generation number, total molecules, the number of generations of stability for considering a regulatory architecture stable (stability gen), stable generations since last instability, the value of the perturbed entity (perturb-value), the phase of the perturbation (perturb-phase), the duration of each perturbation (perturb-time), the frequency of the perturbations (perturb-freq) and the identity of the perturbed entity (perturbed node) were included in the interface.

For simulating changes over time, this model used a combination of deterministic and stochastic functions. The values of each entity/sensor changes at each tick using a deterministic equation: val @ t+1 = val @ t + sum of inputs from all sensors. The change in value contributed by each sensor for a given entity = round((property of entity) x (value of sensor) x (1 - inactive-fraction of sensor) / (weight of regulatory link)). In other words, change = round($k$ x (value of sensor)), where $k$ is a different constant for each sensor of each entity and round indicates rounding to the nearest integer. For positive regulators (link color grey), this change in value was added and for negative regulators (link color black), it was subtracted. The order of operation on the entities/sensors varies with every tick. After every two ticks, only about half the number of each entity/sensor was kept using a random number generator to simulate dilution and random partitioning upon cell division.

The value of each entity/sensor was plotted relative to the most abundant entity/sensor. This profile at each time point can be considered as the 'phenotype' of the system. These scaled values are also used to depict each entity/sensor in the regulatory architecture at each time point.

## Acknowledgements

The author thanks Thomas Kocher, Pierre-Emanuel Jabin, Todd Cooke, Charles Delwiche, and Karen Carleton for discussions; and Thomas Kocher, Pierre-Emanuel Jabin, and members of the Jose Lab for comments on the manuscript. This work was supported in part by National Institutes of Health Grant R01GM124356, National Science Foundation Grant 2120895, and FSRA from UMD to A.M.J.

The author declares no competing interests.

## Additional information

### Funding

| Funder | Grant reference number | Author |
| --- | --- | --- |
| National Institutes of Health | R01GM124356 | Antony M Jose |
| National Science Foundation | 2120895 | Antony M Jose |

The funders had no role in study design, data collection and interpretation, or the decision to submit the work for publication.

### Author contributions

Antony M Jose, Conceptualization, Writing - original draft, Writing - review and editing

### Author ORCIDs

Antony M Jose ⓘ http://orcid.org/0000-0003-1405-0618

Reviewer #1 (Public Review): https://doi.org/10.7554/eLife.92093.3.sa1
Author response https://doi.org/10.7554/eLife.92093.3.sa2

## Additional files

### Supplementary files

• Supplementary file 1. Parameters that generate steady states for the 26 simplest HRAs.

• Supplementary file 2. Behavior of Entity-Sensor-Property systems that have some persistent architecture after 250 generations.

• Supplementary file 3. The timing of *C. elegans* cell divisions along the germline.

• Supplementary file 4. Behavior of regulatory architectures that incorporate the developmental time of *C. elegans*.

• MDAR checklist

### Data availability

The current manuscript is a computational study, so no data have been generated for this manuscript. Modeling code is available at https://github.com/AntonyJose-Lab/Jose_2023, copy archived at *Jose, 2023*.

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
