## [Editor Report · eLife assessment]

This **useful** manuscript explores conditions for epigenetic inheritance by studying the stability of simple network models to permanent and transient perturbations. A novel aspect of the study is that it unifies non-genetic inheritance phenomena across cell divisions of unicellular organisms and in the germline of multicellular organisms. However, the models studied are more a collection of vignettes of numerical studies than a systematic study, therefore the evidence presented remains **incomplete**. As a first step towards building a more systematic theoretical framework, this work will be of interest to colleagues in the field of epigenetic inheritance.

---

## [Referee Report · Reviewer #1 (Public Review)]

The author studies a family of models for heritable epigenetic information, with a focus on enumerating and classifying different possible architectures. The key aspects of the paper are:

- Enumerate all 'heritable' architectures for up-to 4 constituents.

- A study of whether permanent ("genetic") or transient ("epigenetic") perturbations lead to heritable changes

- Enumerated the connectivity of the "sequence space" formed by these heritable architectures

- Incorporating stochasticity, the authors explore stability to noise (transient perturbations)

- A connection is made with experimental results on C elegans.

The study is timely, as there is a renewed interest in the last decade in non-genetic, heritable heterogeneity (e.g., from single-cell transcriptomics). Consequently, there is a need for a theoretical understanding of the constraints on such systems. There are some excellent aspects of this study: for instance, the attention paid to how one architecture "mutates" into another. Unfortunately, the manuscript as a whole does not succeed in formalising nor addressing any particular open questions in the field. Aside from issues in presentation and modelling choices (detailed below), it would benefit greatly from a more systematic approach rather than the vignettes presented.

## Terminology

The author introduces a terminology for networks of interacting species in terms of "entities" and "sensors" -- the former being nodes of a graph, and the latter being those nodes that receive inputs from other nodes. In the language of directed graphs, "entities" would seem to correspond to vertices, and "sensors" those vertices with positive indegree and outdegree. Unfortunately, the added benefit of redefining accepted terminology from the study of graphs and networks is not clear.

## Model

The model seems to suddenly change from Figure 4 onwards. While the results presented here have at least some attempt at classification or statistical rigour (i.e. Fig 4 D), there are suddenly three values associated with each entity ("property step, active fraction, and number"). Furthermore, the system suddenly appears to be stochastic. The reader is left unsure what has happened, especially after having made the effort to deduce the model as it was in Figs 1 through 3. No respite is to be found in the SI, either, where this new stochastic model should have been described in sufficient detail to allow one to reproduce the simulation.

## Perturbations

Inspired especially by experimental manipulations such as RNAi or mutagenesis, the author studies whether such perturbations can lead to a heritable change in network output. While this is naturally the case for permanent changes (such as mutagenesis), the author gives convincing examples of cases in which transient perturbations lead to heritable changes. Presumably, this is due the the underlying multistability of many networks, in which a perturbation can pop the system from one attractor to another.

Unfortunately, there appears to be no attempt at a systematic study of outcomes, nor a classification of when a particular behaviour is to be expected. Instead, there is a long and difficult-to-read description of numerical results that appear to have been sampled at random (in terms of both the architecture and parameter regime chosen). The main result here appears to be that "genetic" (permanent) and "epigenetic" (transient) perturbations can differ from each other -- and that architectures that share a response to genetic perturbation need not behave the same under an epigenetic one. This is neither surprising (in which case even illustrative evidence would have sufficed) nor is it explored with statistical or combinatorial rigour (e.g. how easy is it to mistake one architecture for another? What fraction share a response to a particular perturbation?)

As an additional comment, many of the results here are presented as depending on the topology of the network. However, each network is specified by many kinetic constants, and there is no attempt to consider the robustness of results to changes in parameters.

## DNA analogy

At two points, the author makes a comparison between genetic information (i.e. DNA) and epigenetic information as determined by these heritable regulatory architectures. The two claims the author makes are that (i) heritable architectures are capable of transmitting "more heritable information" than genetic sequences, and (ii) that, unlike DNA, the connectivity (in the sense of mutations) between heritable architectures is sparse and uneven (i.e. some architectures are better connected than others).

In both cases, the claim is somewhat tenuous -- in essence, it seems an unfair comparison to consider the basic epigenetic unit to be an "entity" (e.g., an entire transcription factor gene product, or an organelle), while the basic genetic unit is taken to be a single base-pair. The situation is somewhat different if the relevant comparison was the typical size of a gene (e.g., 1 kb).

---

## [Author Response]

The following is the authors’ response to the current reviews.

**Public Reviews:**

**Reviewer #1 (Public Review):**
The author studies a family of models for heritable epigenetic information, with a focus on enumerating and classifying different possible architectures. The key aspects of the paper are:Enumerate all 'heritable' architectures for up-to 4 constituents.A study of whether permanent ("genetic") or transient ("epigenetic") perturbations lead to heritable changesEnumerated the connectivity of the "sequence space" formed by these heritable architecturesIncorporating stochasticity, the authors explore stability to noise (transient perturbations)A connection is made with experimental results on C elegans.The study is timely, as there is a renewed interest in the last decade in non-genetic, heritable heterogeneity (e.g., from single-cell transcriptomics). Consequently, there is a need for a theoretical understanding of the constraints on such systems. There are some excellent aspects of this study: for instance, the attention paid to how one architecture "mutates" into another. Unfortunately, the manuscript as a whole does not succeed in formalising nor addressing any particular open questions in the field. Aside from issues in presentation and modelling choices (detailed below), it would benefit greatly from a more systematic approach rather than the vignettes presented.

Despite being foundational, this work was systematic in that (1) for the simple architectures modeled using ordinary differential equations (ODEs) with continuity assumptions, parameters that support steady states were systematically determined for each architecture and then every architecture was explored using genetic changes exhaustively, although epigenetic perturbations were not examined exhaustively because of their innumerable variety; and (2) for the more realistic modeling of architectures as Entity-Sensor-Property systems, the behavior of systems with respect to architecture as well as parameter space that lead to particular behaviors (persistence, heritable epigenetic change, etc.) was systematically explored. A more extensive exploration of parameter space that also includes the many ways that the interaction between any two entities/nodes could be specified using an equation is a potentially ever-expanding challenge that is beyond the scope of any single paper.

Specific aspects that remain to be addressed include the application of multiple notions of heritability to real networks of arbitrary size, considering different types of equations for change of each entity/node, and classifying different behavioral regimes for different sets of parameters.

The key contribution of the paper is an articulation of the crucial questions to ask of any regulatory architecture in living systems rather than the addressing of any question that a field has recognized as ‘open’. Specifically, through the exhaustive listing of small regulatory architectures that can be heritable and the systematic analysis of arbitrary Entity-Sensor-Property systems that more realistically capture regulatory architectures in living systems, this work points the way to constrain inferences after experiments on real living systems. Currently, most experimental biologists engaged in reductionist approaches and some systems biologists examining the function or prevalence of network motifs do not explicitly constrain their models for heritability or persistence. It is hoped that this paper will raise awareness in both communities and lead to more constrained models that minimize biases introduced by incomplete knowledge of the network, which is always the case when analyzing living systems.

TerminologyThe author introduces a terminology for networks of interacting species in terms of "entities" and "sensors" -- the former being nodes of a graph, and the latter being those nodes that receive inputs from other nodes. In the language of directed graphs, "entities" would seem to correspond to vertices, and "sensors" those vertices with positive indegree and outdegree. Unfortunately, the added benefit of redefining accepted terminology from the study of graphs and networks is not clear.

The Entities-Sensors-Property (ESP) framework is based on underlying biology and not graph theory, making an ESP system not entirely equivalent to a network or graph, which is much less constrained. The terms ‘entity’, ‘sensor’, and ‘property’ were defined and justified in a previous paper (Jose, J R. Soc. Interface, 2020). While nodes of a network can be parsed arbitrarily and the relationship between them can also be arbitrary, entities and sensors are molecules or collections of molecules that are constrained such that the sensors respond to changes in particular properties of other entities and/or sensors. When considered as digraphs, sensors can be seen as vertices with positive indegree and outdegree. The ESP framework can be applied across any scale of organization in living systems and this specific way of parsing interactions also discretizes all changes in the values of any property of any entity. In short, ESP systems are networks, but not all networks are ESP systems. Therefore, the results of network theory that remain applicable for ESP systems need further investigation.

The key utility of the ESP framework is that it is aligned with the development of mechanistic models for the functions of living systems while being consistent with heredity. In contrast, widely analyzed networks like protein-interaction networks, signaling networks, gene regulatory networks, etc., are not always constrained using these principles.

ModelThe model seems to suddenly change from Figure 4 onwards. While the results presented here have at least some attempt at classification or statistical rigour (i.e. Fig 4 D), there are suddenly three values associated with each entity ("property step, active fraction, and number"). Furthermore, the system suddenly appears to be stochastic. The reader is left unsure what has happened, especially after having made the effort to deduce the model as it was in Figs 1 through 3. No respite is to be found in the SI, either, where this new stochastic model should have been described in sufficient detail to allow one to reproduce the simulation.

The Supplementary Information section titled ‘Simulation of simple ESP systems’ provides the requested detailed information and revisions to the writing provide the biologically grounded justification for parsing interacting regulators as ESP systems.

PerturbationsInspired especially by experimental manipulations such as RNAi or mutagenesis, the author studies whether such perturbations can lead to a heritable change in network output. While this is naturally the case for permanent changes (such as mutagenesis), the author gives convincing examples of cases in which transient perturbations lead to heritable changes. Presumably, this is due the the underlying multistability of many networks, in which a perturbation can pop the system from one attractor to another.Unfortunately, there appears to be no attempt at a systematic study of outcomes, nor a classification of when a particular behaviour is to be expected. Instead, there is a long and difficult-to-read description of numerical results that appear to have been sampled at random (in terms of both the architecture and parameter regime chosen). The main result here appears to be that "genetic" (permanent) and "epigenetic" (transient) perturbations can differ from each other -- and that architectures that share a response to genetic perturbation need not behave the same under an epigenetic one. This is neither surprising (in which case even illustrative evidence would have sufficed) nor is it explored with statistical or combinatorial rigour (e.g. how easy is it to mistake one architecture for another? What fraction share a response to a particular perturbation?)As an additional comment, many of the results here are presented as depending on the topology of the network. However, each network is specified by many kinetic constants, and there is no attempt to consider the robustness of results to changes in parameters.

The systematic study of all arbitrary regulatory architectures is beyond the scope of this paper and, indeed, beyond the scope of any one paper. Nevertheless 225,000 arbitrary Entity-Sensor-Property systems were systematically explored and collections of parameters that lead to different behaviors provided (e.g., 78,285 are heritable). These ESP systems more closely mimic regulation in living systems than the coupled ODE-based specification of change in a regulatory architecture.

The example questions raised here are not only difficult to answer, but subjective and present a moving target for future studies. One, ‘how easy is it to mistake one architecture for another?’. Mistaking one architecture for another clearly depends on the number of different types of experiments one can perform on an architecture and the resolution with which changes in entities can be measured to find distinguishing features. Two, ‘What fraction share a response to a particular perturbation?’. ‘Sharing a response’ also depends on the resolution of the measurement after perturbation.

DNA analogyAt two points, the author makes a comparison between genetic information (i.e. DNA) and epigenetic information as determined by these heritable regulatory architectures. The two claims the author makes are that (i) heritable architectures are capable of transmitting "more heritable information" than genetic sequences, and (ii) that, unlike DNA, the connectivity (in the sense of mutations) between heritable architectures is sparse and uneven (i.e. some architectures are better connected than others).In both cases, the claim is somewhat tenuous -- in essence, it seems an unfair comparison to consider the basic epigenetic unit to be an "entity" (e.g., an entire transcription factor gene product, or an organelle), while the basic genetic unit is taken to be a single base-pair. The situation is somewhat different if the relevant comparison was the typical size of a gene (e.g., 1 kb).

Considering every base being the unit of stored information in the DNA sequence results in the maximal possible storage capacity of a genome of given length. Any other equivalence between entity and units within the genome (e.g., 1 kb gene) will only reduce the information stored in the genome.

Nevertheless, the claim was modified to say that the information content of an ESP system can [italics added] be more extensive than the information content of the genome. This accounts for the possibility of an organism that has an inordinately large genome such that maximal information that can be stored in a particular genome sequence exceeds that stored in a particular configuration of all the contents in a cell.

I thank the reviewer for providing further explanation of this misunderstanding in the second round of review, which helps draw future readers to the sections in the paper that discusses this important point (also see response to Recommendations for the authors).

**Recommendations for the authors:**

**Reviewer #1 (Recommendations For The Authors):**
I thank the author for their efforts in replying to the comments. I have updated my review accordingly; in particular, I have:(1) Removed my complaint that Heritability is nowhere defined(2) Removed issues with the presentation of the ODE model in the supplementary information.

I thank the reviewer for raising these issues and acknowledging the improvements made.

However, given that the manuscript is broadly unchanged from the initial one, many of my prior comments remain justified. Some key points:(1) The manuscript continues to be difficult to read, for the same reasons as I mentioned when reviewing the paper previously.(2) The utility of the "ESP" formalism is still unclear.As the author notes, continuous ODEs are of course an idealisation of a system with discrete copy number.However, discussing this is standard fare in any textbook dealing with chemical dynamics and stochastic processes -- see, for instance, the standard textbook by van Kampen.This seems little reason to reject ODEs and implement a poorly defined formalism/simulation scheme.(3) The author claims that many questions raised are "beyond the scope of this study". Indeed, answering all of these questions are beyond the scope of any one study. However, as I initially wrote, the paper would be much stronger if it focused on a particular problem rather than the many vignettes depicted.

The broad scope of this foundational paper necessitates addressing many issues, which may make it a difficult read for some readers. I hope that future work where each paper focuses on one of the aspects raised here will enable the extensive treatment of limited scope as suggested by the reviewer.

The utility of ODEs is much appreciated and was indeed a computationally efficient way of exploring the vast space of regulatory architectures. As stated in the response to the public reviews, the Entity-Sensors-Property framework provides a biologically grounded way of parsing interacting regulators. This approach is aligned with the development of mechanistic models for the functions of living systems while being consistent with heredity. In contrast, widely analyzed networks like protein-interaction networks, signaling networks, gene regulatory networks, etc., are not always constrained using these principles.

On a final note, on the subject of the comparison with DNA:Perhaps I have misunderstood something. I simply meant that comparing the "maximal information" with 4 HRAs (12.45 bits) is certainly more than the "maximal information" with 4 basepairs (8 bits), but definitely less than the "maximal information" for four 1-kb genes (4^(4000) combinations, so 8000 bits...)Perhaps the author means that the growth in information of HRAs is faster than exponential. If so, that should be shown and then remarked on.For this reason, I maintain my comment that the comparison is tenuous.

This issue was addressed once in the results section and again in the discussion section.

The results section states that “The combinatorial growth in the numbers of HRAs with the number of interactors can thus provide vastly more capacity for storing information in larger HRAs compared to that afforded by the proportional growth in longer genomes.”

The discussion section states that “Despite imposing heritability, regulated non-isomorphic directed graphs soon become much more numerous than unregulated non-isomorphic directed graphs as the number of interactors increase (125 vs. 5604 for 4 interactors, Table 1). With just 10 interactors, there are >3x1020 unregulated non-isomorphic directed graphs [60] and HRAs are expected to be more numerous. This tremendous variety highlights the vast amount of information that a complex regulatory architecture can represent and the large number of changes that are possible despite sparsity of the change matrix (Fig. 3).”

Thus, indeed as the reviewer surmises, the combinatorial explosion in information of HRAs with increases in interacting entities is faster than the proportional growth in information of genome sequence with increases in length.

In summary, I thank the reviewers and editors for their help in improving the paper and would like to make the current manuscript the Version of Record.

The following is the authors’ response to the original reviews.

**Public Reviews:**

**Reviewer #1 (Public Review):**
The author studies a family of models for heritable epigenetic information, with a focus on enumerating and classifying different possible architectures. The key aspects of the paper are:Enumerate all 'heritable' architectures for up to 4 constituents.A study of whether permanent ("genetic") or transient ("epigenetic") perturbations lead to heritable changes.Enumerated the connectivity of the "sequence space" formed by these heritable architectures.-Incorporating stochasticity, the authors explore stability to noise (transient perturbations). - A connection is made with experimental results on C elegans.The study is timely, as there has been a renewed interest in the last decade in nongenetic, heritable heterogeneity (e.g., from single-cell transcriptomics). Consequently, there is a need for a theoretical understanding of the constraints on such systems. There are some excellent aspects of this study: for instance:The attention paid to how one architecture "mutates" into another, establishing the analogue of a "sequence space" for network motifs (Fig 3).The distinction is drawn between permanent ("genetic") and transient ("epigenetic") perturbations that can lead to heritable changes.The interplay between development, generational timescales, and physiological time (as in Fig. 5).

I thank the reviewer for highlighting these aspects of the work.

The manuscript would be very interesting if it focused on explaining and expanding these results. Unfortunately, as a whole, it does not succeed in formalising nor addressing any particular open questions in the field. Aside from issues in presentation and modelling choices (detailed below), it would benefit greatly from a more systematic approach rather than the vignettes presented.

This first paper is foundational and therefore cannot be expected to solve all aspects of the problem of heredity. The work was nevertheless systematic in that (1) for the simple architectures modeled using ordinary differential equations (ODEs) with continuity assumptions, parameters that support steady states were systematically determined for each architecture and then every architecture was explored using genetic changes exhaustively, although epigenetic perturbations were not examined exhaustively because of their wide variety; and (2) for the more realistic modeling of architectures as Entity-Sensor-Property systems, the behavior of systems with respect to architecture as well as parameter space that lead to particular behaviors (persistence, heritable epigenetic change, etc.) was systematically explored. A more extensive exploration of parameter space that also includes the many ways that the interaction between any two entities/nodes could be specified using an equation is a potentially ever-expanding challenge that is beyond the scope of any single paper (see response to additional comments below).

Specific aspects that remain to be addressed include the application of multiple notions of heritability to real networks of arbitrary size, considering different types of equations for change of each entity/node, and classifying different behavioral regimes for different sets of parameters. As is evident from this list of combinatorial possibilities, the space to be explored is vast and beyond the scope of this foundational paper.

The key contribution of the paper is an articulation of the crucial questions to ask of any regulatory architecture in living systems rather than the addressing of any question that a field has recognized as ‘open’. Specifically, through the exhaustive listing for small regulatory architectures that can be heritable and the systematic analysis of arbitrary Entity-Sensor-Property systems that more realistically capture regulatory architectures in living systems, this work points the way to constrain inferences after experiments on real living systems. Currently, most experimental biologists engaged in reductionist approaches and some systems biologists examining the function or prevalence of network motifs do not explicitly constrain their models for heritability or persistence. It is hoped that this paper will raise awareness in both communities and lead to more constrained models that minimize biases introduced by incomplete knowledge of the network, which is always the case when analyzing living systems.

TerminologyThe author introduces a terminology for networks of interacting species in terms of "entities" and "sensors" -- the former being nodes of a graph, and the latter being those nodes that receive inputs from other nodes. In the language of directed graphs, "entities" would seem to correspond to vertices, and "sensors" those vertices with positive indegree and outdegree. Unfortunately, the added benefit of redefining accepted terminology from the study of graphs and networks is not clear.

The Entities-Sensors-Property (ESP) framework is based on underlying biology and not graph theory, making an ESP system not entirely equivalent to a network or graph, which is much less constrained. The terms ‘entity’, ‘sensor’, and ‘property’ were defined and justified in a previous paper (Jose, J R. Soc. Interface, 2020). While nodes of a network can be parsed arbitrarily and the relationship between them can also be arbitrary, entities and sensors are molecules or collections of molecules that are constrained such that the sensors respond to changes in particular properties of other entities and/or sensors. When considered as digraphs, sensors can be seen as vertices with positive indegree and outdegree. The ESP framework can be applied across any scale of organization in living systems and this specific way of parsing interactions also discretizes all changes in the values of any property of any entity. In short, ESP systems are networks, but not all networks are ESP systems. Therefore, the results of network theory that remain applicable for ESP systems need further investigation. This justification is now repeated in the paper.

The key utility of the ESP framework is that it is aligned with the development of mechanistic models for the functions of living systems while being consistent with heredity. In contrast, widely analyzed networks like protein-interaction networks, signaling networks, gene regulatory networks, etc., are not always constrained using these principles. In addition, the language of digraphs where sensors can be seen as vertices with positive indegree and outdegree has been also added to aid readers who are familiar with graph theory.

HeritabilityThe primary goal of the paper is to analyse the properties of those networks that constitute "heritable regulatory architectures". The definition of heritability is not clearly stated anywhere in the paper, but it appears to be that the steady-state of the network must have a non-zero expression of every entity. As this is the heart of the paper, it would be good to have the definition of heritable laid out clearly in either the main text or the SI.

I have now defined the term as used in this paper early, which is indeed as surmised by the reviewer simply the preservation of the architecture and non-zero levels of all entities. I have also highlighted additional notions of heredity that are possible, which will be the focus of future work. These can range from precise reproduction of the concentration and the localization of every entity to a subset of the entities being reproduced with some error while the rest keep varying from generation to generation (as illustrated in Fig. 2 of Jose, BioEssays, 2018). Importantly, it is currently unclear which of these possibilities reflects heredity in real living systems.

ModelAs described in the supplementary, but not in the main text, the author first chooses to endow these networks with simple linear dynamics; something like ∂tx→=Ax−Tx, where the vector *x* is the expression level of each entity, *A* has the structure of the adjacency matrix of the directed graph, and *T* is a diagonal matrix with positive entries that determines the degradation or dilution rate of each entity. From a readability standpoint, it would greatly aid the reader if the long list of equations in the SI were replaced with the simple rule that takes one from a network diagram to a set of ODEs.

I have abridged the description by eliminating the steady state expression for every HRA as suggested and simply pointed to the earlier version of the paper for those readers who might prefer the explicit derivations of these simple expressions. An overview is now provided for going from any network diagram to a set of ODEs.

The implementation of negative regulation is manifestly unphysical if the "entities" represent the expression level of, say, gene products. For instance, in regulatory network E, the value of the variable z can go negative (for instance, if the system starts with z = and y=0, and x > 0).

Negative values for any entity were avoided in simulations by explicitly setting all such values to zero. This constraint has been added as a note in the section describing the equations for the change of each node/entity in each regulatory network. Specifically, the levels of each entity/sensor was set to zero during any time step when the computed value for that entity/sensor was less than zero. This bounding of the function allows for any approach to zero while avoiding negative values. I apologize for the omission of this constraint from the supplemental material in the last submission. This constraint was used in all the simulations and therefore this change does not affect any of the results presented. In this way, it is ensured that the presence of negative regulation does not lead to negative values.

Formally, the promotion or inhibition of an entity or sensor can be modeled using any function that is either increasing (for promotion) or decreasing (for inhibition). This diversity of possibilities is one of the challenges that prevents exhaustive exploration of all functions. In fact, the use of ODEs after assuming a continuous function is an idealization that facilitates understanding of general principles but is not in keeping with the discreteness of entities or step changes in their values (amount, localization, etc.) observed in living systems. Other commonly used continuous functions include Hill functions for the rate of production of y given as x^n^/(k + x^n^) for x activating y, which increases to ~1 as x increases, or given as k/(k + x^n^) for x inhibiting y, which decreases to ~0 as x increases. Increasing values of ‘n’ result in steeper sigmoidal curves. In reality, levels of all entities/sensors are expected to be discretized by measurement in living systems and the form of the function for any regulation needs empirical measurement in vivo (see response to comment below).

The model seems to suddenly change from Figure 4 onwards. While the results presented here have at least some attempt at classification or statistical rigour (i.e. Fig 4 D), there are suddenly three values associated with each entity ("property step, active fraction, and number"). Furthermore, the system suddenly appears to be stochastic. The reader is left unsure of what has happened, especially after having made the effort to deduce the model as it was in Figs 1 through 3. No respite is to be found in the SI, either, where this new stochastic model should have been described in sufficient detail to allow one to reproduce the simulation.

While ODEs are easier to simulate and understand, they are less realistic as explained above. I have now added more explanation justifying the need for the subsequent simulation of Entity-Sensor-Property systems. I have also expanded the information provided for each aspect of the model (previously outlined in Fig. 4A and detailed within the code) in a Supplementary Information section titled ‘Simulation of simple ESP systems’.

PerturbationsInspired especially by experimental manipulations such as RNAi or mutagenesis, the author studies whether such perturbations can lead to a heritable change in network output. While this is naturally the case for permanent changes (such as mutagenesis), the author gives convincing examples of cases in which transient perturbations lead to heritable changes. Presumably, this is due the the underlying mutlistability of many networks, in which a perturbation can pop the system from one attractor to another.Unfortunately, there appears to be no attempt at a systematic study of outcomes, nor a classification of when a particular behaviour is to be expected. Instead, there is a long and difficult-to-read description of numerical results that appear to have been sampled at random (in terms of both the architecture and parameter regime chosen). The main result here appears to be that "genetic" (permanent) and "epigenetic" (transient) perturbations can differ from each other -- and that architectures that share a response to genetic perturbation need not behave the same under an epigenetic one. This is neither surprising (in which case even illustrative evidence would have sufficed) nor is it explored with statistical or combinatorial rigour (e.g. how easy is it to mistake one architecture for another? What fraction share a response to a particular perturbation?)

The systematic study of all arbitrary regulatory architectures is beyond the scope of this paper and, as stated earlier, beyond the scope of any one paper. Nevertheless 225,000 arbitrary Entity-Sensor-Property systems were systematically explored and collections of parameters that lead to particular behaviors provided (e.g., 78,285 are heritable). These ESP systems more closely mimic regulation in living systems than the coupled ODE-based specification of change in a regulatory architecture.

The example questions raised here are not only difficult to answer, but subjective and present a moving target for future studies. One, ‘how easy is it to mistake one architecture for another?’. Mistaking one architecture for another clearly depends on the number of different types of experiments one can perform on an architecture and the resolution with which changes in entities can be measured to find distinguishing features. Two, ‘What fraction share a response to a particular perturbation?’. ‘Sharing a response’ also depends on the resolution of the measurement of entities after perturbation.

As an additional comment, many of the results here are presented as depending on the topology of the network. However, each network is specified by many kinetic constants, and there is no attempt to consider the robustness of results to changes in parameters.

The interpretations presented are conservative determinations of heritability based on the topology of the architecture. In other words, architectures that can be heritable for some set of parameters. Of course, parameter sets can be found that make any regulatory architecture not heritable. As stated earlier, exploring all parameters for even one architecture is beyond the scope of a single study because of the infinitely many ways that the interaction between any two entities can be specified.

DNA analogyAt two points, the author makes a comparison between genetic information (i.e. DNA) and epigenetic information as determined by these heritable regulatory architectures. The two claims the author makes are that (i) heritable architectures are capable of transmitting "more heritable information" than genetic sequences, and (ii) that, unlike DNA, the connectivity (in the sense of mutations) between heritable architectures is sparse and uneven (i.e. some architectures are better connected than others).In both cases, the claim is somewhat tenuous -- in essence, it seems an unfair comparison to consider the basic epigenetic unit to be an "entity" (e.g., an entire transcription factor gene product, or an organelle), while the basic genetic unit is taken to be a single base-pair. The situation is somewhat different if the relevant comparison was the typical size of a gene (e.g., 1 kb).

Considering every base being the unit of stored information in the DNA sequence results in the maximal possible storage capacity of a genome of given length. Any other equivalence between entity and units within the genome (e.g., 1 kb gene) will only reduce the information stored in the genome.

Nevertheless, the claim has been modified to say that the information content of an ESP system can [italics added] be more extensive than the information content of the genome. This accounts for the possibility of an organism that has an inordinately large genome such that maximal information that can be stored in a particular genome sequence exceeds that stored in a particular configuration of all the contents in a cell.

**Reviewer #2 (Public Review):**
Summary:This manuscript uses an interesting abstraction of epigenetic inheritance systems as partially stable states in biological networks. This follows on previous review/commentary articles by the author. Most of the molecular epigenetic inheritance literature in multicellular organisms implies some kind of templating or copying mechanisms (DNA or histone methylation, small RNA amplification) and does not focus on stability from a systems biology perspective. By contrast, theoretical and experimental work on the stability of biological networks has focused on unicellular systems (bacteria), and neglects development. The larger part of the present manuscript (Figures 1-4) deals with such networks that could exist in bacteria. The author classifies and simulates networks of interacting entities, and (unsurprisingly) concludes that positive feedback is important for stability. This part is an interesting exercise but would need to be assessed by another reviewer for comprehensiveness and for originality in the systems biology literature. There is much literature on "epigenetic" memory in networks, with several stable states and I do not see here anything strikingly new.

The key utility of the initial part of the paper is the exhaustive enumeration of all small heritable regulatory architectures. The implications for the abundance of ‘network motifs’ and more generally any part of a network proposed to perform a particular function is that all such parts need to be compatible with heredity. This principle is generally not followed in the literature, resulting in incomplete networks being interpreted as having motifs or modules with autonomous function. Therefore, while the need for positive feedback for stability is indeed obvious, it is not consistently applied by all. For example, the famous synthetic circuit ‘the repressilator’ (Elowitz and Leibler, “A synthetic oscillatory network of transcriptional regulators”, Nature, 2000), which is presented as an example of ‘rational network design’, has three transcription factors that all sequentially inhibit the production of another transcription factor in turn forming a feedback loop of inhibitory interactions. Therefore, the contributions of the factors that promote the expression of each entity is unknown and yet essential for heritability. The comprehensive listing of the heritable regulatory architectures that are simple provide the basis for true synthetic biology where the contributing factors for observed behavior of the network are explicitly considered only after constraining for heredity. Using this principle, the minimal autonomous architecture that can implement the repressilator is the HRA ‘Z’ (Fig. 1).

An interesting part is then to discuss such networks in the framework of a multicellular organism rather than dividing unicellular organisms, and Figure 5 includes development in the picture. Finally, Figure 6 makes a model of the feedback loops in small RNA inheritance in *C. elegans* to explain differences in the length of inheritance of silencing in different contexts and for different genes and their sensitivity to perturbations. The proposed model for the memory length is distinct from a previously published model by Karin et al. (ref 49).

I thank the reviewer for appreciating this aspect of the paper.

Strengths:A key strength of the manuscript is to reflect on conditions for epigenetic inheritance and its variable duration from the perspective of network stability.

I thank the reviewer for appreciating the importance of the overall topic.

Weaknesses:I found confusing the distinction between the architecture of the network and the state in which it is. Many network components (proteins and RNAs) are coded in the genome, so a node may not disappear forever.

I have added language to clarify the many states of a network versus its architecture (also illustrated in Fig. 4 for ESP systems). Even loss of expression below a threshold can lead to permanent loss if there is not sufficient noise to induce re-expression. For example, consider the simple case of a transcription factor that binds to its own promoter, requiring 10 molecules for the activation of the promoter and thus production of more of the same transcription factor. If an epigenetic change (e.g., RNA interference) reduces the levels to fewer than 10 molecules and if the noise in the system never results in the numbers of the transcription factor increasing beyond 10, the transcription factor has been effectively lost permanently. In this way, reduction of a regulator can lead to permanent change despite the presence of the DNA. Many papers in the field of RNA silencing in *C. elegans* have provided strong experimental evidence to support this assertion.

From the Supplementary methods, the relationship between two nodes seems to be all in the form of dx/dt = Kxy . Y, which is just one way to model biological reactions. The generality of the results on network architectures that are heritable and robust/sensitive to change is unclear. Other interactions can have sigmoidal effects, for example. Is there no systems biology study that has addressed (meta)stability of networks before in a more general manner?

Indeed, the relationship between any two entities can in principle be modeled using any function. Extensive exploration of the behavior of any regulatory architecture – even the simplest ones – require simplifications. For example, early work by Stuart Kauffman explored Boolean networks (see ref. 10 in the paper for history and extensive explanations). However, allowing all possible ways of specifying the interactions between components of a network makes analysis both a computational and conceptual challenge.

Why is auto-regulation neglected? As this is a clear cause of metastable states that can be inherited, I was surprised not to find this among the networks.

Auto-regulation in the sense of some molecule/entity ultimately leading to the production of more of itself is present in every heritable regulatory architecture. Specifically, all auto-regulatory loops rely on a sequence of interactions between two or more kinds of molecules. For example, a transcription factor (TF) binding to the promoter of its own gene sequence, resulting in the production of more TF protein is a positive feedback loop that relies on many interacting factors (transcription, translation, nuclear import, etc.) and can be considered as ‘auto-regulation’ as it is sometimes referred to in the literature. In this sense, every HRA (A through Z) includes ‘auto-regulation’ or more appropriately positive feedback loops. For example, in the HRA ‘A’, x ‘auto-regulates’ itself via y.

I did not understand the point of using the term "entity-sensor-property". Are they the same networks as above, now simulated in a computer environment step by step (thus allowing delays)?

Please see response to the other reviewer regarding the need for the Entity-SensorProperty framework and how it is distinct from generic networks. Briefly, the ODE-based simple networks, while easy to analyze, are not realistic because of the assumptions of continuity. In contrast ESP systems are more realistic with measurement discretizing changes in property values as is expected in real living systems.

The final part applies the network modeling framework from above to small RNA inheritance in *C. elegans*. Given the positive feedback, what requires explanation is how fast the system STOPs small RNA inheritance. A previous model (Karin et al., ref. 49) builds on the fact that factors involved in inheritance are in finite quantity hence the different small RNAs "compete" for amplification and those targeting a given gene may eventually become extinct.The present model relies on a simple positive feedback that in principle can be modulated, and this modulation remains outside the model. A possibility is to add negative regulation by factors such as HERI-1, that are known to limit the duration of the silencing.The duration of silencing differs between genes. To explain this, the author introduces again outside the model the possibility of piRNAs acting on the mRNA, which may provide a difference in the stability of the system for different transcripts. At the end, I do not understand the point of modeling the positive feedback.

The previous model (Karin et al., Cell Systems, 2023) can describe populations of genes that are undergoing RNA silencing but cannot explain the dynamics of silencing particular genes. Furthermore, this model also cannot explain cases of effectively permanent silencing of genes that have been reported (e.g., Devanapally et al., Nature Communications, 2021 and Shukla et al., Current Biology, 2021). Finally, the observations of susceptibility to, recovery from, and even resistance to trans silencing (e.g., Fig. 5a in Devanapally et al., Nature Communications, 2021) require an explanation that includes modulation of the HRDE-1-dependent positive feedback loop that maintains silencing across generations.

The specific qualitative predictions regarding the relationship between piRNA-mediated regulation genome-wide and HRDE-1-dependent silencing of a particular gene across generations could guide the discovery of potential regulators of heritable RNA silencing. The equations (4) and (5) in the paper for the extent of modulation needed for heritable epigenetic change provide specific quantitative predictions that can be tested experimentally in the future. I have also revised the title of the section to read ‘Tuning of positive feedback loops acting across generations can explain the dynamics of heritable RNA silencing in *C. elegans*’ to emphasize the above points.

From the initial analysis of abstract networks that do not rely on templating, I expected a discussion of possible examples from non-templated systems and was a little surprised by the end of the manuscript on small RNAs.

The heritability of any entity relies on regulatory interactions regardless of whether a templated mechanism is also used or not. For example, DNA replication relies on the interactions between numerous regulators, with only the sequence being determined by the template DNA. The field of small RNA-mediated silencing facilitates analysis of epigenetic changes at single-gene resolution (Chey and Jose, Trends in Genetics, 2022). It is therefore likely to continue to provide insights into heritable epigenetic changes and how they can be modulated. Unfortunately, there are currently no known cases of epigenetic inheritance where the role of any templated mechanism has been conclusively excluded. Future research will improve our understanding of epigenetic states and their modulation in terms of changes in positive feedback loops as proposed in this study and potentially lead to the discovery of such mechanisms that act entirely independent of any template-dependent entity.

**Recommendations for the authors:**

I thank the reviewers for their specific suggestions to improve the paper.

**Reviewer #1 (Recommendations For The Authors):**
The paper has many long paragraphs that attempt to explain results, make illustrations, and give intuition. Unfortunately, these are difficult to read. It would aid the reader greatly if these were, say, converted into cartoons (even if only in the SI), or made more accessible in some other way.

I agree with the importance of making the material accessible to readers in multiple ways. I have now added a figure with schematics in the SI titled ‘Illustrations of key concepts’ (new Fig. S2), which collects concepts that are relevant throughout the paper and might aid some readers.

The bulk of the supplementary is currently a collection of elementary mathematics results: to whit, pages 26 to 33 of the combined manuscript carry no more information than a quick description of the general model and the diagrams in Fig 1. Similarly, pages 34 to 39 (non-zero dilution rate), and pages 39 through 58 (response to permanent changes) each express a trivial mathematical point that is more than sufficiently made with one illustrative example.

I agree with the reviewer and have condensed these pages as suggested. I have added a pointer to the earlier version as containing further details for the readers who might prefer the explicit listing of these equations.

Overall, the paper appears to be a collection of numerical results obtained from different models, united by uncertain terminology that is not fully defined in this paper. The most promising aspects of the paper lie either in (a) combinatorially complete enumeration of all regulatory architectures, or (b) relating experimental manipulations in *C. elegans* to possible underlying regulatory architectures. Focusing on one or the other might improve the readability of the paper.

The two sections of the paper are complementary and when presented together help with the integration of concepts rather than the siloed pursuit of theory versus experimental analysis. When this work was presented at meetings before submission, it was clear that different researchers appreciated different aspects. This divergence is also apparent in the two reviews, with each reviewer appreciating different aspects. I have repeated the definitions and justifications from the earlier paper (Jose, J R Soc Interface, 2020) to provide a more fluid transition between the two complementary sections of the paper. Knowing both sides could aid in the development of models that are not only consistent with measurable quantities (e.g., anything that can be considered an entity) but are also logically constrained (e.g., entities matched with sensors while avoiding any entities that do not have a source of production – i.e., avoiding nodes with indegree = 0).

However, having said that many results of these types are well-known in models of regulatory networks, and it is unclear what precisely warrants the new framework that the author is proposing. Indeed, it would be good to understand in what way the framework here is novel, and how it is distinguished from prior studies of regulatory networks.

The key novelty of the work is the consideration of heritability for any regulation. With the explicit definition of the heritability for a regulatory architecture and the acknowledgement that there can be more than one notion of heredity, this paper now sets the foundation for examining many real networks in this light. I hope that the added justifications for the current framework in the revised paper strengthen these arguments. Future literature reviews on networks in general and how they address heritability or persistence will better define the prevalence of these considerations. Currently, most experimental biologists engaged in reductionist approaches and some systems biologists examining the function or prevalence of network motifs do not explicitly constrain their models for heritability or persistence. It is hoped that this work will raise awareness in both communities and lead to more constrained models that acknowledge incomplete knowledge of the network, which is always the case when analyzing living systems.

**Reviewer #2 (Recommendations For The Authors):**
Minor points/claritypage 1 line 57: "transgenerational waveforms that preserve form and function" is unclear.

This phrase was expanded upon in a previous paper (Jose, BioEssays, 2020). I have now added more explanation in this paper for completeness. The section now reads ‘For example, the localization and activity of many kinds of molecules are recreated in successive generations during comparable stages [1-3]. These recurring patterns can change throughout development such that following the levels and/or localizations of each kind of molecule over time traces waveforms that return in phase with the similarity of form and function across generations [2].’

page 7 line 3-6: the sentence has an ambiguous structure.

I have now edited this long sentence to read as follows: ‘For systematic analysis, architectures that could persist for ~50 generations without even a transient loss of any entity/sensor were considered HRAs. Each HRA was perturbed (loss-of-function or gain-of-function) after five different time intervals since the start of the simulation (i.e., phases). The response of each HRA to such perturbations were compared with that of the unperturbed HRA.’

page 9 lines 25-27: the sentence is convoluted: are you defining epigenetic inheritance?

I have simplified this sentence describing prior work by others (Karin et al., Cell Systems, 2023) and moved a clause to the subsequent sentence. This section now reads: ‘Recent considerations of competition for regulatory resources in populations of genes that are being silenced suggest explanations for some observations on RNA silencing in *C. elegans* [49]. Specifically, based on Little’s law of queueing, with a pool of M genes silenced for an average duration of T, new silenced genes arise at a rate λ that is given by M = λT’. I have also provided more context by preceding this section with: ‘Although the release of shared regulators upon loss of piRNA-mediated regulation in animals lacking PRG-1 could be adequate to explain enhanced HRDE-1-dependent transgenerational silencing initiated by dsRNA in prg-1(-) animals, such a competition model alone cannot explain the observed alternatives of susceptibility, recovery and resistance (Fig. 6A).’

page 13 lines 51-53. This last sentence of the discussion is ambiguous/unclear.

I have now rephrased this sentence to read: ‘This pathway for increasing complexity through interactions since before the origin of life suggests that when making synthetic life, any form of high-density information storage that interacts with heritable regulatory architectures can act as the ‘genome’ analogous to DNA.’

Figure 2: the letters in the nodes are hard to read; the difference between full and dotted lines in the graphs also.

I have enlarged the nodes and widened the gap in the dotted lines to make them clearer. I have also similarly edited Fig. 1 and Fig. S3 to Fig. S9.